# SCAR-6 elncRNA locus epigenetically regulates *PROZ* and modulates coagulation and vascular function

Gyan Ranjan [ID] [1,2], Paras Sehgal [1,2], Vinod Scaria [ID] [1,2,5,6 ✉] & Sridhar Sivasubbu [ID] [1,2,3,4 ✉]

## Abstract

In this study, we characterize a novel lncRNA-producing gene locus that we name *Syntenic Cardiovascular Conserved Region-Associated lncRNA-6 (scar-6)* and functionally validate its role in coagulation and cardiovascular function. A 12-bp deletion of the *scar-6* locus in zebrafish (*scar-6$^{gib007Δ12/Δ12}$*) results in cranial hemorrhage and vascular permeability. Overexpression, knockdown and rescue with the *scar-6* lncRNA modulates hemostasis in zebrafish. Molecular investigation reveals that the *scar-6* lncRNA acts as an enhancer lncRNA (elncRNA), and controls the expression of *prozb*, an inhibitor of *factor Xa*, through an enhancer element in the *scar-6* locus. The *scar-6* locus suppresses loop formation between *prozb* and *scar-6* sequences, which might be facilitated by the methylation of CpG islands via the *prdm14-PRC2* complex whose binding to the locus might be stabilized by the *scar-6* elncRNA transcript. Binding of *prdm14* to the *scar-6* locus is impaired in *scar-6$^{gib007Δ12/Δ12}$* zebrafish. Finally, activation of the PAR2 receptor in *scar-6$^{gib007Δ12/Δ12}$* zebrafish triggers *NF-κB*-mediated endothelial cell activation, leading to vascular dysfunction and hemorrhage. We present evidence that the *scar-6* locus plays a role in regulating the expression of the coagulation cascade gene *prozb* and maintains vascular homeostasis.

**Keywords** Zebrafish; Endothelial Cell; Cardiovascular; Hemorrhage; Synteny lncRNA
**Subject Categories** Chromatin, Transcription & Genomics; RNA Biology; Vascular Biology & Angiogenesis

## Introduction

Long non-coding RNAs (lncRNAs) have been investigated for their functions and have been found to regulate protein-coding genes in *cis* or *trans* in various biological conditions (Kopp and Mendell, 2018; Gil and Ulitsky, 2020). To date, only a handful of lncRNA genes have been functionally characterized using model organisms, such as mice or zebrafish, to understand complex regulatory networks in disease and development (Ranjan et al, 2021;

Sauvageau et al, 2013; Han et al, 2018; Sehgal et al, 2021; Kurian et al, 2016; Hosono et al, 2017). Numerous lncRNAs, including *MALAT1, Caren, MANTIS, MHRT, TUG1*, and *Pnky*, have been shown to be physiologically crucial for organ development, maturation, and function (Cremer et al, 2019; Sato et al, 2021; Leisegang et al, 2017; Lewandowski et al, 2020; Ramos et al, 2015). These lncRNAs provide compelling evidence of their significant roles in regulating key biological processes, underscoring their importance in maintaining proper organ physiology (Oo et al, 2022). However, genomic comparisons of lncRNAs based on nucleotide sequences across vertebrates have shown low or no sequence conservation, posing a significant challenge in identifying orthologs (Hezroni et al, 2015). Multiple studies have shown that lncRNAs exhibit alternative conservation modes independent of their sequence (Diederichs, 2014; Ranjan et al, 2021). lncRNAs arising from conserved syntenic loci and thus exhibiting functional conservation is one such mode that is frequently used to study lncRNA orthologs. Syntenic loci are evolutionarily conserved regions on a chromosome that exhibit a similar order of homologous genes across different species, primarily defined around protein-coding genes. Several lncRNA orthologs proximal to homologous protein-coding genes have been identified using the conserved synteny approach in different model organisms (Ranjan et al, 2021). The availability of large catalog datasets of lncRNAs from zebrafish has recently opened avenues for understanding the lncRNA-mediated regulatory aspects of conserved developmental and morphological pathways across vertebrates (Hu et al, 2018). Multiple investigations have previously identified many syntenically conserved lncRNAs in zebrafish that have shown functional importance. *Tie1-AS, PUNISHER, ALIEN, MALAT1, TERMINATOR, Durga*, and *PU.1 AS* are some lncRNAs arising from syntenic loci that show the presence of functional orthologs between humans and zebrafish (Ranjan et al, 2021; Kurian et al, 2016; Sarangdhar et al, 2017; Li et al, 2010; Wei et al, 2014; Wu et al, 2019; Ulitsky et al, 2011).

In addition, uncovering the functional modalities of these lncRNA locus has been a central focus of the field.

LncRNA loci are categorized based on their mechanisms of action. Some function through their mature RNA molecules, interacting with proteins, RNA, or DNA to regulate cellular processes (e.g., *VEAL2, GATA6-AS, Tie1-AS, lncREST*) (Sehgal et al, 2021; Neumann et al, 2018; Li et al, 2010; Statello et al, 2024). Others act through the transcription process, serving as

[1]CSIR Institute of Genomics and Integrative Biology, Mathura Road, Delhi 110024, India. [2]Academy of Scientific and Innovative Research (AcSIR), Ghaziabad 201002, India. [3]Vishwanath Cancer Care Foundation, Mumbai, India. [4]Dr. D. Y Patil Medical College, Hospital and Research Centre, Pune, India. [5]Present address: Vishwanath Cancer Care Foundation, Mumbai, India. [6]Present address: Dr. D. Y Patil Medical College, Hospital and Research Centre, Pune, India. ✉E-mail: drvinod@gmail.com; sridhar@igib.in

bidirectional promoters or transcription and splicing regulators (e.g., *ARIN, Handsdown*) (Santoro et al, 2013; Ritter et al, 2019). In addition, some loci operate exclusively through their DNA elements, functioning as chromatin, enhancer, or transcription factor binding regions (e.g., *PVT1, linc-p21, Lockd, Meteor*) (Cho et al, 2018; Groff et al, 2016; Rom et al, 2019; Gil et al, 2023). The RNAs produced from these loci are often termed as enhancer RNAs (eRNAs) or enhancer-associated lncRNAs (elncRNAs). They play significant roles in biological processes like vascular function and development. eRNAs are classified into two categories based on their features and polyadenylation (poly-A) (Lam et al, 2014; Natoli and Andrau, 2012; St Laurent et al, 2015; Li et al, 2016). The first class consists of eRNAs that are bidirectional, single-exonic, non-polyadenylated, and short in size. These RNAs are produced from enhancers and are often considered transcriptional noise or features of enhancers. The second class includes eRNAs or elncRNAs that exhibit characteristics similar to lncRNAs, such as unidirectional transcription, multi-exonic structure, length greater than 500 bp and polyadenylation (Koch et al, 2011; Lam et al, 2014; Hou et al, 2019). These elncRNA exhibits higher enhancer activity due to post-transcriptional processing of these RNAs (Ørom et al, 2010; Gil and Ulitsky, 2018; Arner et al, 2015; Tan and Marques, 2022). elncRNAs produced from these loci are known to interact with transcription factors (TFs) and stabilize their binding at the locus, thereby mediating transcription. In addition, eRNAs are involved in phase condensation and methylation via m6A modification, which are critical for active enhancer functions (Mattick et al, 2023; Lee et al, 2021). The involvement of eRNAs in phase condensates has been reported, underscoring their importance in cellular processes. Moreover, multiple studies have indicated that targeting these elncRNAs can have therapeutic potential, highlighting their relevance in disease contexts (Zhang et al, 2019; Allou et al, 2021; Cajigas et al, 2018; Katsushima et al, 2024; Fatima et al, 2019; Sun et al, 2014).

The cardiovascular system comprises a complex network of tissues that orchestrates blood circulation and other essential components throughout the body. Recent advancements have identified numerous proteins crucial for the maintenance and functionality of vascular and cardiac tissues (Srivastava and Olson, 2000; Trimm and Red-Horse, 2022). Transcriptional regulation within these tissues is pivotal for maintaining systemic homeostasis, and disruptions in these regulatory networks can lead to pathological conditions such as coronary artery disease (CAD), stroke, and heart failure (Spielmann et al, 2022; Marsman et al, 2013; Anene-Nzelu et al, 2021; Mathew and Sivasubbu, 2022; Park et al, 2013). In addition, recent studies have underscored the significance of various coagulation factor proteins in cardiovascular function (Zhao and Schooling, 2018; Loeffen et al, 2012). These enzymes, which contain the serine protease domain, are integral not only to coagulation but also to mediating pro-inflammatory signaling via the activation of protease-activated receptors (PAR1, 2, 3, and 4) (Posma et al, 2019; Heuberger and Schuepbach, 2019). Activating these receptors by clotting factors such as thrombin and FXa can induce endothelial cell activation, leading to vascular dysfunction(Rabiet et al, 1996; Bono et al, 2000). However, the transcriptional regulation of these proteins remains poorly understood, partly due to the lethality of knockout models for these factors. Zebrafish have recently emerged as a valuable model system for studying coagulation and cardiovascular genetics due to their

conserved coagulation cascade and cardiovascular gene networks (Liu et al, 2014; Hu et al, 2017; Weyand et al, 2019). Unlike mouse models, zebrafish can tolerate severe defects during early development, facilitating research on these pathways (Dewerchin et al, 2000; Liu et al, 2014; Cui et al, 1996; Hu et al, 2017; Jalbert et al, 1998). Furthermore, emerging evidence suggests that elncRNAs regulate enhancers or promoters of neighboring protein-coding genes involved in cardiovascular function. Examples of such elncRNAs include *AIRN, Chaserr, Handsdown, Pcdhα-as, Upperhand, GAL10-ncRNA, Haunt, PVT1,* and *Fxt*. These elncRNAs play crucial roles in modulating gene expression and maintaining cardiovascular homeostasis (Ali and Grote, 2020; Yin et al, 2015; Ritter et al, 2019; Rom et al, 2019; Canzio et al, 2019; Latos et al, 2012; Han et al, 2019; Furlan et al, 2018; Cho et al, 2018; Hosen et al, 2018).

In this study, we genetically probed the role of previously uncharacterized syntenic lncRNA locus *scar-6* in zebrafish and for the first time, provided provided evidence that the *scar-6* lncRNA locus functions as an enhancer, producing an elncRNA transcript that collaboratively regulates the *prozb* gene. It also controls coagulation and vascular function in vivo. The *scar-6* locus consists of an enhancer that modulates *prozb* expression via binding of the *prdm14-PRC2* complex at the locus which is stabalized by the *scar-6* elncRNA. The *prdm14-PRC2* complex modulates the methylation of nearby CpG island, affecting the CTCF binding and enhancer (*scar-6*)-promoter (*prozb*) looping. Moreover, the upregulation of *prozb* in the mutant *scar-6* zebrafish resulted in the activation of the PAR-2 receptor, causing upregulation of *ICAM1* and *VCAM1* expression, leading to vascular dysfunction. This study puts forward the role of *scar-6* elncRNA locus in modulating coagulation and vascular function via controlling the expression of the *prozb* gene.

## Results

### Identification of syntenic lncRNA genes associated with the cardiovascular system

We used a systematic approach to identify syntenic lncRNA genes in the cardiovascular system. Using the genome map in the ZFLNC database (Hu et al, 2018), we listed out 5 neighboring protein-coding genes of all syntenic lncRNA genes in zebrafish (337 lncRNA genes) and obtained a total of 571 neighboring protein-coding genes. We compiled a comprehensive list of cardiovascular system-associated genes by querying the Zfin database with keywords such as "heart" (2531 genes), "blood vessels" (1122 genes), and "endothelium" (287 genes) (Bradford et al, 2022). This process resulted in a consolidated list comprising 2854 protein-coding genes. We then overlapped the 571 protein-coding genes proximal to syntenic lncRNAs with the cardiovascular-associated gene lists (2854 genes). We found 291 overlapping protein-coding genes in proximity to 98 syntenic lncRNA genes. To enhance the likelihood of identifying syntenic lncRNA genes involved in the cardiovascular system, we excluded those with only one adjacent protein-coding gene. We focused on retaining syntenic lncRNA genes that had multiple neighboring protein-coding genes associated with cardiovascular functions. We also excluded lncRNAs that were single exonic transcripts and retained multiexonic

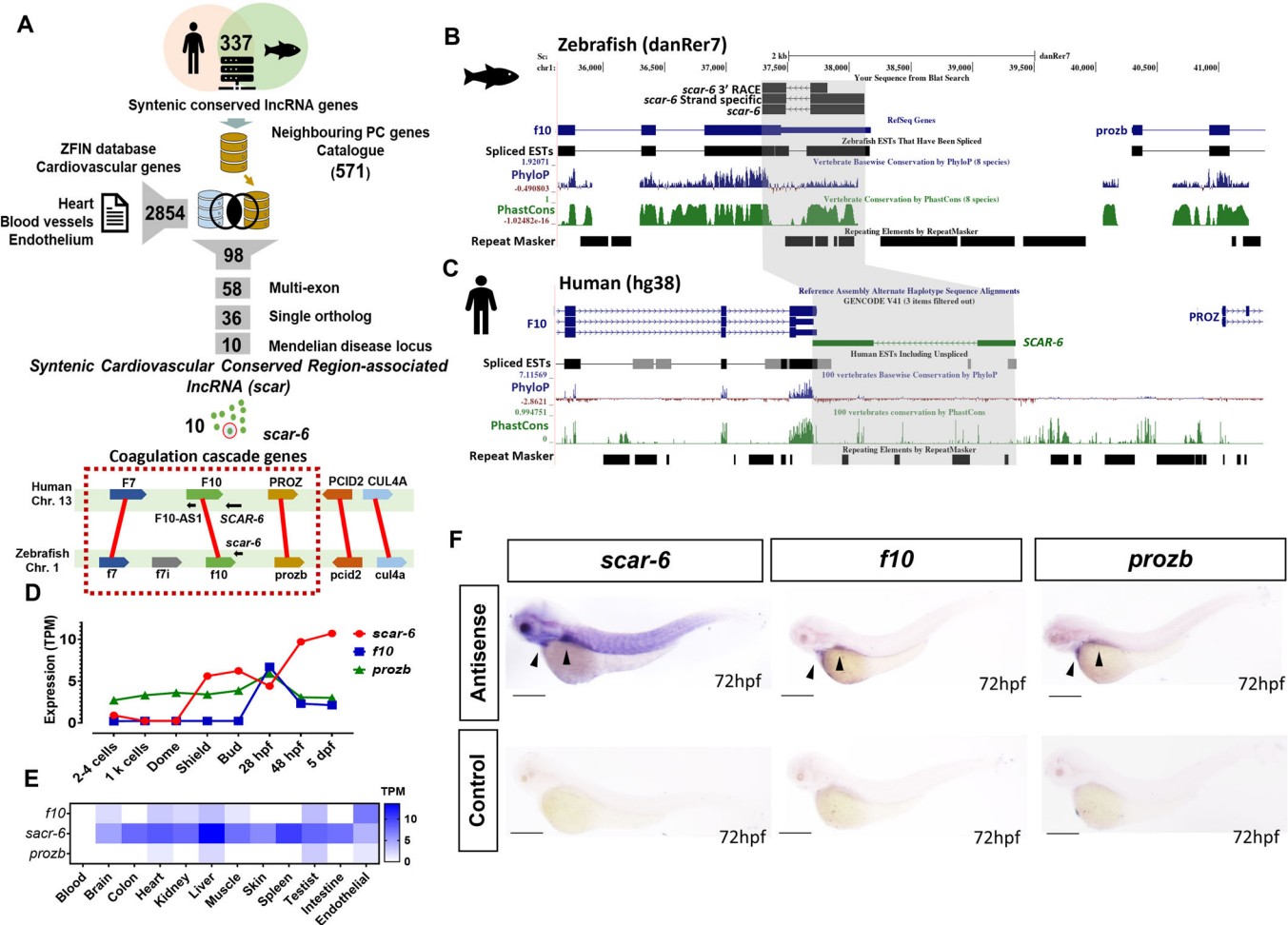

**Figure 1. Syntenic cardiovascular conserved associated region - 6 (scar-6) is a novel conserved lncRNA gene.**

(A) Schematic of pipeline used for subset selection of syntenic lncRNAs with its neighboring protein-coding genes associated in the cardiovascular system. Overview of conserved syntenic block depiction of *scar-6* locus between human and zebrafish. (B, C) UCSC genome browser snapshot of *scar-6* locus in zebrafish and human. Strand-specific PCR and 3'RACE confirmed the transcript to be antisense to *f10* in zebrafish. (D) Expression profile of *scar-6* transcript (in TPM) across 8 different developmental stages of zebrafish using publically available RNA-seq data. (E) Expression profile of *scar-6* transcript (in TPM) across 12 different tissues using publically available RNA-seq data. (F) Whole mount in situ hybridization expression analysis of *scar-6, f10*, and *prozb* transcripts of zebrafish in 3 dpf embryo. Sense probes were used as controls for *f10* and *prozb*. No probe control was used for *scar-6*. Black arrow head denotes expression in liver and heart. 4× magnification scale bar = 500 μm. Source data are available online for this figure.

lncRNAs (58 lncRNA gene) (Andersson et al, 2014). In addition, we selected lncRNAs that had single predicted human ortholog transcripts to avoid any potential confusion (36 lncRNA gene) and further refine our selection process by prioritizing candidate lncRNAs based on the Mendelian disease associations of their neighboring genes and got 10 potential syntenic lncRNAs with possible involvement in the cardiovascular system (Appendix Table S1). We named the selected lncRNA genes as *Syntenic Cardiovascular Conserved Region-Associated lncRNA (scar)* (Fig. 1A; Appendix Fig. S1).

### scar-6 is a novel uncharacterized conserved lncRNA gene

We selected *scar-6* lncRNA gene as a candidate for functional validation as it belongs to a highly conserved syntenic block comprising the genes involved in the coagulation cascade, and no

lncRNA has been previously characterized for its functional association with coagulation and the cardiovascular system, making it a novel target for our investigation. The *scar-6* lncRNA locus produces a bi-exonic polyadenylated lncRNA (Fig. EV1A). It has a transcript length of 632 bp observed in a synteny block on chromosome 1 of zebrafish and arises antisense to 3' UTR of the *coagulation factor X (f10)* gene and upstream to *protein zb (prozb)* gene (Fig. 1B). Strand-specific PCR and 3'-rapid amplification of cDNA ends (RACE), and strand specific amplification conform *scar-6* to be an independent antisense transcript to the *f10* gene (Fig. 1B; Appendix Fig. S2). The *scar-6* lncRNA locus has a putative conserved lncRNA gene in humans which also produces bi-exonic novel transcript, antisense to *F10* (ENSG00000283828) (previously named AL137002.2 or *LOC102724474*) (hereafter mentioned as *SCAR-6*). It is present in the human syntenic block on chromosome 13 with length of 2238 bp and antisense to the *F10* gene, overlapping with its

3′ UTR and upstream to *PROZ* gene (Fig. 1C). The transcript shows no sequence conservation with its putative zebrafish orthologue (Appendix Figs. S3–S4). We also computed the non-coding potential of the lncRNAs using bioinformatic algorithms such as the Coding Potential Assessment Tool (CPAT2) (coding probability—0.0003) (Wang et al, 2013) or Coding Potential Calculator (CPC2) (coding probability—0.0192227) (Kang et al, 2017), and found *scar-6* lncRNA to exhibit no coding potential (Fig. EV1B). In addition, sub-cellular investigation for *scar-6* lncRNA and its neighboring genes in zebrafish showed enrichment of *scar-6* and *prozb* expression in the nuclear compartment. In contrast, *f10* showed expression in both the cytoplasmic and nuclear compartments (Fig. EV1C,D). Subsequently, we conducted expression analysis of *scar-6* and adjacent genes by re-analyzing RNA-seq datasets from developmental stages and adult tissues, which are publicly accessible (Appendix Table S2) (Pauli et al, 2012; Yang et al, 2020; Sehgal et al, 2021). We observed *scar-6* to be expressed from the bud stage to 5dpf. It showed a similar expression pattern to *f10* after 28hpf and no primary pattern was observed for *prozb* throughout the developmental stages (Fig. 1D). We observed *scar-6* to be ubiquitously expressed in 11 of the tested 12 zebrafish adult tissues and show higher expression in the liver. The expression of neighboring genes *f10* and *prozb* showed relatively very low expression in selected tissues (Fig. 1E). As both *f10* and *prozb* are expressed and secreted out from the liver (Heinz and Braspenning, 2015; Kemkes-Matthes and Matthes, 1995), we also observed similar expression of *scar-6* lncRNA and its neighboring protein-coding genes in our expression analysis. In situ hybridization at 3 and 5 dpf of zebrafish showed a concordant expression pattern of *scar-6* lncRNA in zebrafish (Fig. 1F; Appendix Fig. S5). Similar tissue expression patterns were also observed in human tissues in the GTEx v8 database for the *SCAR-6* lncRNA gene (Ferraro et al, 2020) (Fig. EV1E). This data indicates that the *scar-6* locus produces an RNA transcript exhibiting typical characteristics of lncRNAs, including non-coding nature, biexonic structure with a poly-A tail, and a length greater than 500 bp.

## The *scar-6* locus modulates vascular and hemostasis function

To understand the function of *scar-6* locus in the coagulation cascade and cardiovascular system, we generated a stable mutant line using the CRISPR-Cas9 system. We designed sgRNAs targeting the 1st and 2nd exon of *scar-6* lncRNA gene in a double transgenic zebrafish gib004Tg(*fli1: EGFP; gata1a:dsRed*) background (Lalwani et al, 2012; Lawson and Weinstein, 2002; Traver et al, 2003). In our initial F$_0$ screening, we observed no phenotype in the cardiovascular system of zebrafish crispants targeted for exon 1. However, we noted cranial hemorrhage in approximately 30% of crispants with the sgRNA targeting the 2nd exon (3′ region) of the *scar-6* gene. Therefore, we selected these mutants for further study (Appendix Fig. S6). We used a systematic approach to generate a stable F$_2$ mutant animal line with a 12-bp deletion in exon 2 at position 594–606 bases of *scar-6* transcript and named it *scar-6*$^{gib007Δ12}$ (detailed in Appendix data section) (Figs. 2A,B and EV2A; Appendix Fig. S7). The *scar-6*$^{gib007Δ12/+}$ animals were checked for morphological or patterning defects in the cardiovascular system till 5dpf in double transgenic background zebrafish (gib004Tg(*fli1: EGFP; gata1a:dsRed*)) under a fluorescent microscope. We observed no significant defects in patterning and formation of blood vessels.

We next in-crossed two *scar-6*$^{gib007Δ12/+}$ animals and observed the progeny displaying genotype in the mendelian ratio (Fig. 2C,D; Appendix Table S3). Upon further investigation, we observed that 22.7% (*p*-value = 0.0023, unpaired two-tail t-test) of the offspring exhibited cranial hemorrhage phenotype starting from 60 hpf and prominent at 72 hpf (3 dpf) (Fig. 2C,E). Subsequently, upon genotyping these hemorrhage animals from three independent in-crosses of *scar-6*$^{gib007Δ12/+}$ animals demonstrated that 96% (86/90) of the animals carried the homozygous mutation (Fig. 2G). This suggests that homozygous mutant animals of *scar-6*$^{gib007Δ12/Δ12}$ exhibit cranial hemorrhage phenotype (Fig. 2F). We also observed the embryos under a confocal microscope and fluorescent microscope to understand the cause of hemorrhage, but we did not find any visible vascular malformations or patterning defects in the mutant zebrafish (Fig. EV2B). To further characterize, we checked for any permeability defect in our mutant by injecting Evans blue dye into the zebrafish's common cardinal vein (CCV). Notably, *scar-6*$^{gib007Δ12/Δ12}$ animals exhibited permeability defects, evidenced by dye leakage between the intersegmental vessels (ISV). This observation strongly indicates that the permeability defect in *scar-6*$^{gib007Δ12/Δ12}$ animals led to hemorrhage (Fig. 2H,I).

Next, we examined the molecular alterations in the zebrafish mutant of the *scar-6* lncRNA. We observed that there was an 8-fold increase in the expression of the upstream neighboring gene *prozb* in the *scar-6*$^{gib007Δ12/Δ12}$ animals when compared to the wild-type counterparts (*p*-value ≤ 0.001, unpaired two-tailed t-test). However, we did not find any significant changes in the expression levels of *f10 and scar-6* when comparing the *scar-6*$^{gib007Δ12/Δ12}$ animals to the wild-type counterparts (Fig. 2J). This implies that the *scar-6* locus functions in a *cis*-regulatory manner, specifically influencing *prozb* expression. The unchanged levels of *scar-6* lncRNA suggest that the 12-bp deletion may not affect its degradation but could potentially alter its secondary structure, thereby disrupting its functional role. Therefore, we examined the secondary structure of the lncRNA with and without the 12-bp deletion using RNAfold web server (Lorenz et al, 2011). We observed a significant alteration in the secondary structure of the lncRNA following the deletion of 12 bp (Fig. EV2C). The observed changes, particularly in the hairpin loop structure, could profoundly impact the *scar-6* lncRNA's function. This structural alteration might affect its ability to interact with target molecules, regulate gene expression, and stabilize transcription factor binding at the regulatory site. These disruptions could render the *scar-6* lncRNA nonfunctional, explaining the observed changes in *prozb* expression.

We further evaluated the hemostasis ability of *scar-6* in the zebrafish. We in-crossed the *scar-6*$^{gib007Δ12/+}$ animals and the embryos were further subjected to a mechanical vascular injury at the cardinal vein (CV) of 3 dpf animals. The occlusion time was recorded blindly, and subsequent genotyping was performed (Clay and Coughlin, 2015). We observed a significant loss of hemostasis ability in both heterozygous (*scar-6*$^{gib007Δ12/+}$) and homozygous (*scar-6*$^{gib007Δ12/Δ12}$) mutants of *scar-6* animals. The mean occlusion time was 40.5 s for heterozygous mutants and 120 s for homozygous mutants, compared to only 24.5 s for the control group (*p*-value < 0.0001, Mann–Whitney *U* test) (Fig. 3A). We also analyzed the hemostasis ability upon overexpression of the wild-type *scar-6* lncRNA in zebrafish and recorded the occlusion time blindly upon mechanical vascular injury at the CV. We injected 3 different concentrations (100 ng/μL, 200 ng/μL, and 500 ng/μL) of the *scar-6*

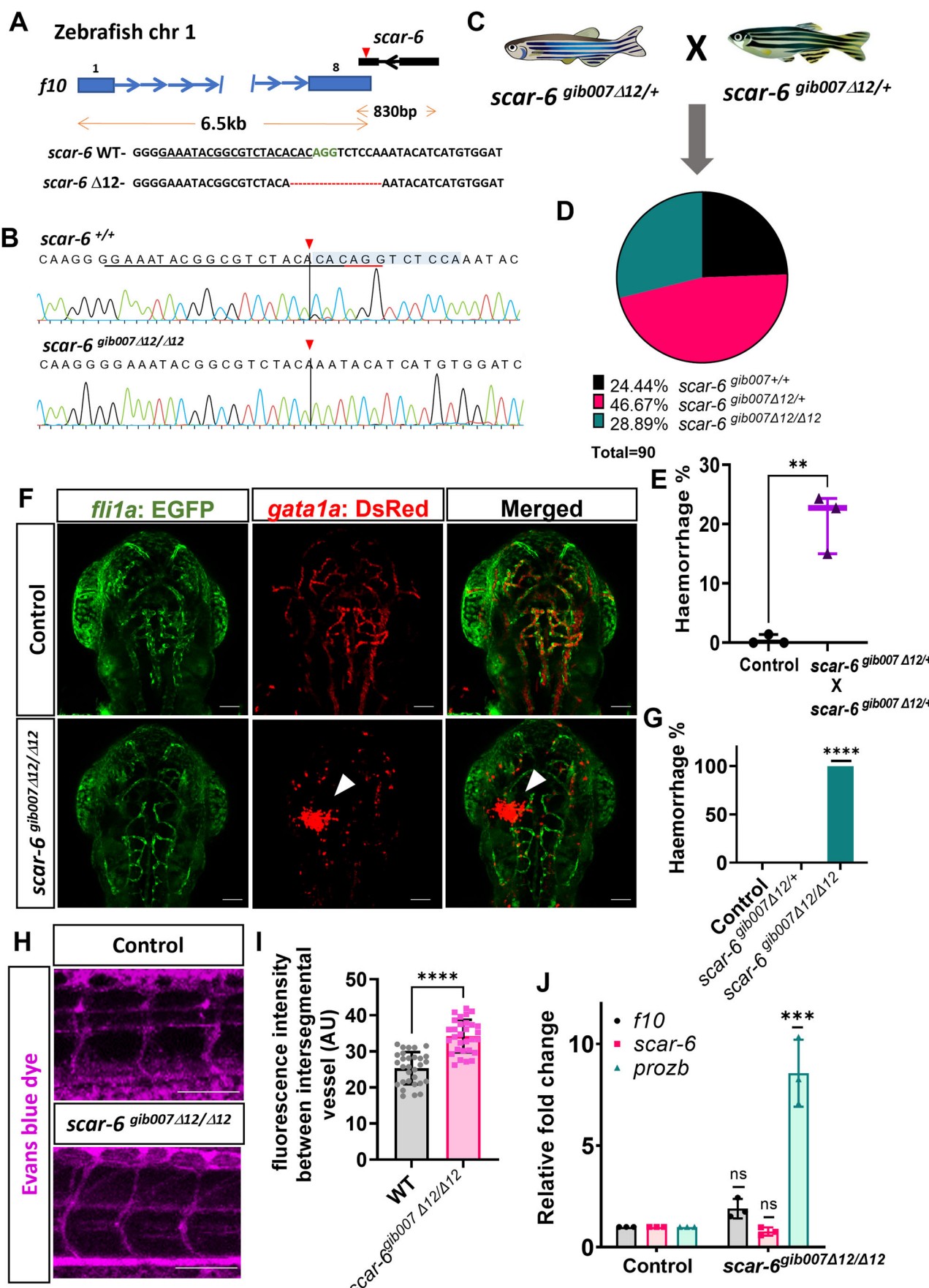

**Figure 2.**  *scar-6*$^{gib007Δ12/Δ12}$ **mutant exhibits permeability defect with haemorrhage phenotype.**

(A) CRISPR-Cas9 mediated editing of *scar-6* locus at exon 2 resulting in a stable 12-bp deletion zebrafish line named *scar-6*$^{gib007Δ12}$. (B) Chromatogram representing the sequence of wild type and homozygous mutant of *scar-6*$^{gib007Δ12}$. (C) Schematic representing in-cross of *scar-6*$^{gib007Δ12/+}$ zebrafish. (D) Pie chart representing mutation segregation of the progeny from in-cross of *scar-6*$^{gib007Δ12/+}$ zebrafish. The genotype of the progeny followed the Mendelian ratio. (E) Box and whisker plot representing the percentage of animals from the in-cross of *scar-6*$^{gib007Δ12/+}$ zebrafish exhibiting haemorrhage phenotype compared to wild-type control animals. Data from 3 independent in-cross of *scar-6*$^{gib007Δ12/+}$ zebrafish are represented by boxes indicating the interquartile range (25th to 75th percentiles), with the horizontal line within each box denoting the median. Whiskers extend to 1.5 times the interquartile range to define the minimum and maximum values, while individual points represent data from each experiment; **$p = 0.0023$ (two-tailed unpaired t-test). (F) Representative image showing the cranial region of 3 dpf zebrafish wild-type gib004Tg(*fli1a*:EGFP;*gata1a*:DsRed) and *scar-6*$^{gib007Δ12/Δ12}$ zebrafish. The white arrowhead denotes haemorrhage in the animals. 40× magnification ; scale bar = 100 μm. (G) Bar plot representation of hemorrhage phenotype associated with genotyped *scar-6*$^{gib007Δ12/+}$ in-crossed zebrafish animals. Data from 3 independent in-cross of *scar-6*$^{gib007Δ12/+}$ zebrafish experiment plotted as mean percentage ± standard deviation. ****$p$-value ≤ 0.0001 (unpaired two-tail t-test). (H) Representative image showing intersegmental vessels (ISV) of 3 dpf wild-type and *scar-6*$^{gib007Δ12/Δ12}$ zebrafish injected with Evans blue dye. 20× magnification; scale bar = 100 μm. (I) Quantitative analysis of vascular permeability in wild-type and *scar-6*$^{gib007Δ12/Δ12}$ zebrafish injected with Evans blue dye. Data of 10 different ROI from 4 individual zebrafish from each wild-type and *scar-6*$^{gib007Δ12/Δ12}$ zebrafish plotted as mean ± standard deviation; ****$p$-value ≤ 0.0001 (Mann–Whitney U). (J) Bar plot representing relative fold change in expression levels of *f10*, *scar-6* and *prozb* between wild-type and *scar-6*$^{gib007Δ12/Δ12}$ zebrafish. The *prozb* expression levels were 8-fold increase in the *scar-6*$^{gib007Δ12/Δ12}$ zebrafish compared to the wild-type. Data from 3 independent biological replicates plotted as mean fold change ± standard deviation; ***$p$-value ≤ 0.001 (unpaired two-tail t-test). Source data are available online for this figure.

in vitro transcribed (IVT) RNA into the single-cell wild-type zebrafish (Appendix Fig. S10A). We observed a significant enhancement of hemostasis in a dose-dependent manner when *scar-6* lncRNA was overexpressed, suggesting its probable role in hemostasis ($p$-value < 0.0001, Mann-Whitney U test) (Fig. 3B).

Knockout models of coagulation cascade genes in mice have been characterized as embryonic lethal (Dewerchin et al, 2000; Cui et al, 1996; Denis et al, 1998). Also, recent studies in zebrafish suggest that knockouts of coagulation cascade genes are embryonic viable but exhibit adult lethality (Liu et al, 2014; Hu et al, 2017; Weyand et al, 2019). Therefore, we investigated the survival rate of *scar-6*$^{gib007Δ12/Δ12}$ zebrafish embryos for 6 months. We observed that the wild-type animals exhibit a normal survival rate of 80%. In contrast, the *scar-6*$^{gib007Δ12/Δ12}$ animals displayed a significant reduction in survival rate, with only 20% of them surviving ($p$-value < 0.0001, log-rank test) (Fig. 3C). We conducted a morphological assessment of the animals during the survival analysis at 30 days post-fertilization (dpf). Notably, we observed that the homozygous mutants exhibited carinal hemorrhage defects of varying severity, mainly categorized as severe, moderate, and minimal (Fig. 3D,E; Appendix Fig. S8).

To exclude the possibility that *f10* disruption contributes to the observed phenotype, we evaluated both mRNA and protein levels of *f10* and found no significant alterations (Fig. 2J; Appendix Fig. S9A). In addition, a rescue experiment by overexpressing *f10* mRNA IVT (100 ng/μL) in both wild-type and *scar-6*$^{gib007Δ12/Δ12}$ animals showed no rescue of the phenotype suggesting no involvement of *f10* in the observed phenotype of *scar-6* mutant (Appendix Fig. S9B,C).

## *scar-6* acts as enhancer lnRNA and exhibits *cis*-regulatory function

We next evaluate the functional modality of the *scar-6* locus. We attempted to rescue the phenotype of *scar-6*$^{gib007Δ12/Δ12}$ animals by injecting wild-type *scar-6* IVT RNA (100 ng/μL) into the progeny of in-crossed *scar-6*$^{gib007Δ12/+}$ animals. We assessed the hemostatic ability of the mutants after rescue with either the wild-type or 12 bp deleted *scar-6* lncRNA IVT RNA. Our results showed that the in-crossed *scar-6*$^{gib007Δ12/+}$ animals could restore hemostatic function with the wild-type copy of *scar-6*, but not with the 12 bp deleted copy (Fig. 3F). In addition, there was no change in the percentage

of animals with the hemorrhage phenotype (Fig. 3G,H) and no molecular change in the expression of *prozb* (Fig. 3I).

To further investigate the role of the *scar-6* lncRNA, we conducted a transient knockdown using an anti-sense splice-blocking morpholino (Appendix Table S4). We observed no prominent cardiovascular phenotype until 5 dpf upon knockdown (Fig. EV3A–C). Although a cranial hemorrhage phenotype was observed in a few animals at 3 dpf, the numbers were not significant (Fig. EV3C,D). In addition, there was no change in the expression levels of *f10* and *prozb* upon *scar-6* lncRNA knockdown or overexpression (Fig. EV3E; Appendix Fig. S10B). However, a significant change in hemostasis efficiency was noted upon knockdown of the *scar-6* lncRNA (Fig. EV3F). These findings indicate that the RNA produced from the *scar-6* locus plays a role in regulating the hemostatic ability of zebrafish.

Taken together these observations suggests the RNA produced from the *scar-6* locus does not appear to impact cranial hemorrhage morphology or significant change in the transcriptional regulation of the *prozb* gene in *scar-6*$^{gib007Δ12/Δ12}$ animals. We speculate that the function of the *scar-6* lncRNA may be context-dependent in its role in controlling cranial hemorrhage and *prozb* regulation. Alternatively, other regulatory elements, possibly enhancer, might be involved in regulating *prozb* expression and cranial hemorrhage in the *scar-6* mutants.

Further we evaluate the presence of a DNA regulatory element as the functional modality of the *scar-6* locus. We observed that the *scar-6* lncRNA locus was overlapping with an annotated enhancer region at the 2nd exon of the lncRNA loci, which is active during the prim-5 and log pec stages of zebrafish (Baranasic et al, 2022) (Fig. 4A; Appendix Fig. S11). We also observed that in humans, the *SCAR-6* loci exhibited an overlap with distal enhancers (known as CRE) as annotated by ENCODE datasets (ENCODE Project Consortium et al, 2020). Moreover, examining chromatin state marks across various human cell lines further confirmed the active regulatory function of the *SCAR-6* locus in a cell line-specific manner (Roadmap Epigenomics Consortium et al, 2015) (Fig. 4B). In addition, we investigated the TF binding sites on the *scar-6* locus, considering both human and zebrafish species and found 68 TF which were showing multiple binding sites and enrichment in both human and zebrafish *scar-6* locus (Appendix Fig. S12). These TF upon gene ontology analyses for biological significance were majorly enriched in embryonic organ morphogenesis, heart

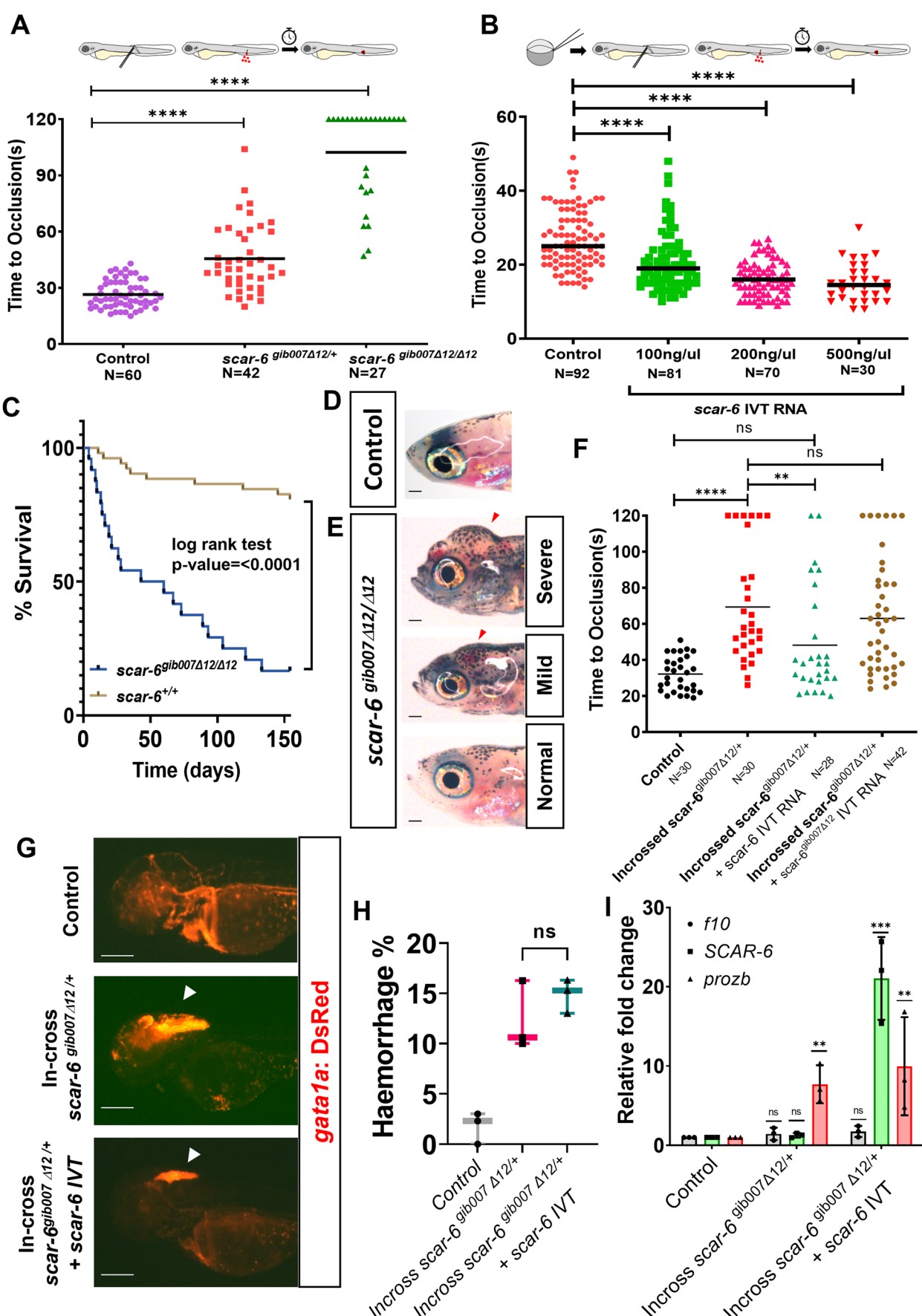

**Figure 3. The scar-6 locus exhibits a functional role in the hemostatic process.**

(A) Coagulation assay plot calculating the time of occlusion (s) in different individual zebrafish in control and progeny of in-cross scar-6$^{gib007\Delta12/+}$ zebrafish performed blinded of the genotype of the animals. Combined data from 3 independent experiments with each point representing the occlusion time of individual zebrafish segregated based on genotype: ****$p < 0.0001$ (Mann-Whitney U). (B) Coagulation assay plot calculating the time of occlusion (s) in different individual zebrafish in control and different concentration of scar-6 IVT (100, 200, 500 ng/µl) injected embryos performed blinded of the injected concentration. Combined data from 3 independent experiments with each point representing the occlusion time of individual zebrafish segregated based on injected concentration; ****$p < 0.0001$ (Mann–Whitney U). (C) Survival curve representing survival percentage of closely monitored control ($n = 52$) and scar-6$^{gib007\Delta12/\Delta12}$ ($n = 24$) zebrafish. $p < 0.0001$ (log-rank test). (D, E) Representative image at 30 dpf zebrafish of control and scar-6$^{gib007\Delta12/\Delta12}$ group closely monitored for survival. The red arrowhead indicates haemorrhage and cranial edema in scar-6$^{gib007\Delta12/\Delta12}$ mutant animals. 1.5× magnification scale bar = 200 µm. (F) Coagulation assay plot calculating the time of occlusion (s) in different individual zebrafish in control, in-cross progeny of scar-6$^{gib007\Delta12/+}$, in-cross progeny of scar-6$^{gib007\Delta12/+}$ injected with WT scar-6 IVT RNA(100 ng/µl) and in-cross progeny of scar-6$^{gib007\Delta12/+}$ injected with scar-6$^{gib007\Delta12}$ IVT RNA(100 ng/µl) performed blinded. Each point representing the occlusion time of individual zebrafish segregated based on injected concentration; ns, not significant, **$p < 0.01$, ****$p < 0.0001$ (Mann–Whitney U). (G) Representative image of the cranial region of zebrafish at 3 dpf of control wild-type, in-cross progeny of scar-6$^{gib007\Delta12/+}$ and in-cross progeny of scar-6$^{gib007\Delta12/+}$ injected with WT scar-6 IVT RNA(100 ng/µl). The white arrowhead denotes haemorrhage in the animals. 2.5× magnification scale bar = 200 µm. (H) Box and whisker plot representing the percentage of animals from control wild-type, in-cross of scar-6$^{gib007\Delta12/+}$ zebrafish and, in-cross of scar-6$^{gib007\Delta12/+}$ + scar-6 IVT RNA (100 ng/µl) exhibiting haemorrhage phenotype. Data from 3 independent experiments are represented by boxes indicating the interquartile range (25th to 75th percentiles), with the horizontal line within each box denoting the median. Whiskers extend to 1.5 times the interquartile range to define the minimum and maximum values, while individual points represent data from each experiment; ns, not significant (two-tailed unpaired t-test). (I) Relative fold change expression of scar-6, f10, and prozb when compared between control wild-type, scar-6$^{gib007\Delta12/\Delta12}$ animals and scar-6$^{gib007\Delta12/\Delta12}$ + scar-6 IVT RNA (100 ng/µl) animals. Data from 3 independent biological replicates plotted as mean fold change ± standard deviation; ns, not significant; **$p < 0.01$, ***$p$-value ≤ 0.001 (two-tailed unpaired t-test). Source data are available online for this figure.

looping, cranial development, and other developmental processes (Appendix Fig. S12D). To further validate the locus as an enhancer, we performed an enhancer assay by cloning the zebrafish scar-6 and human SCAR-6 locus DNA before a minimal promoter E1b followed by an eGFP reporter and injected it in the single-cell embryo of zebrafish. We observed eGFP signals in the cardiac region of injected animals for both zebrafish scar-6 and human SCAR-6 enhancer plasmid. eGFP signal in the trunk region of zebrafish was also observed in animals injected with human SCAR-6 enhancer plasmid (Fig. 4C). Furthermore, we also performed the reporter enhancer assay in human cell culture models using psiCHEK-2 plasmid. We cloned the human SCAR-6 and zebrafish scar-6 locus DNA upstream to the SV40 promoter which drives the renilla luciferase (hRluc) and used the firefly luciferase (hluc) driven by HSV-TK promoter to normalize the expression. The plasmids were transfected in HHL-17, HEK297T, HUVEC/tert2, and HepG2 cell lines. We observed that the human SCAR-6 locus exhibits an enhancer function in HepG2 and HUVEC/hTert2 cells whereas it exhibited a repressor function in the cell lines HHL-17 and HEK297T cell lines after normalization. This was also concordant to the epigenome chromatin marks of HepG2 and HUVEC cells showing enhancer mark and other cell-lines showing quiescent marks (Roadmap Epigenomics Consortium et al, 2015) (Fig. 4A). Interestingly we did not find any change in the expression with the plasmid containing zebrafish scar-6 DNA locus in human cell cultures (Fig. 4D). The enhancer assay in zebrafish, luciferase assay in cell lines, and associated chromatin marks on the scar-6 lncRNA gene body in both human and zebrafish collectively suggest that the scar-6 locus also functions as an enhancer element which regulates prozb expression.

## The scar-6 locus regulates prozb expression through the prdm14-PRC2 complex

Next, we conducted an experiment to assess the functional impact of transcription factors binding at the scar-6 locus using the CRISPR-dCas9-KRAB complex along with the same sgRNA targeting the 12 bp mutation region. Notably, we observed a similar hemorrhage phenotype in ~30% of the animals that

survived beyond 3 dpf after injection with dCas9-KRAB-sgscar-6. In contrast, the control group injected with dCas9-KRAB only showed no such hemorrhage phenotype. These findings support the hypothesis that the observed hemorrhage phenotype is specifically attributed to the precises editing of the enhancer at the scar-6 locus or locus-specific inhibition of transcription factor binding at the enhancer present on scar-6 locus (Fig. EV4).

Throughout the development process, it has been observed that distal enhancers play a crucial role in achieving cell type specificity by interacting with promoters through sub-TAD looping. Therefore, we next investigated if the enhancer element in the scar-6 lncRNA locus and prozb gene promoter exhibit any potential looping between them. Using the Hi-C data from human HepG2 cell lines at 10 kb resolution obtained from the ENCODE project (ENCODE Project Consortium et al, 2020; Wang et al, 2018) and zebrafish brain Hi-C data at 5 kb resolution (Yang et al, 2020), we observed the presence of sub-TAD looping interactions between the scar-6 enhancer locus and the prozb gene promoter in both human and zebrafish genomes indicating conserved physical interactions between scar-6/prozb (Fig. EV5A; Appendix Figs. S13–S14). To validate the looping, we further performed a chromosome conformation capture (3-C) experiment followed by qPCR specific to the scar-6 enhancer locus and the prozb gene promoter. We observed that the scar-6$^{gib007\Delta12/\Delta12}$ animals exhibit an increase in the subTAD-looping when compared to the wild-type animals (Fig. 4E). As CTCF-cohesin plays a major role in intra-TAD associations, we also looked into the CTCF binding site at the scar-6 locus using the publicly available CTCF-ChIP seq data of zebrafish; (Pérez-Rico et al, 2020) We observed that at 24 hpf, there is a presence of CTCF binding at the scar-6 and the prozb loci (Fig. EV5B). We further validated the CTCF occupancy at the scar-6 locus by performing ChIP-qRT-PCR. Compared to wild-type animals, we observed an enhanced CTCF occupancy at the scar-6 locus in scar-6$^{gib007\Delta12/\Delta12}$ animals (Fig. EV5C). This suggests that the upregulation of prozb expression in scar-6$^{gib007\Delta12/\Delta12}$ animals is due to an increase in CTCF occupancy causing an increase in the looping between the scar-6 enhancer and prozb promoter.

Next, we examined the 68 TF which were showing multiple binding sites at the scar-6 locus obtained previously and we observed

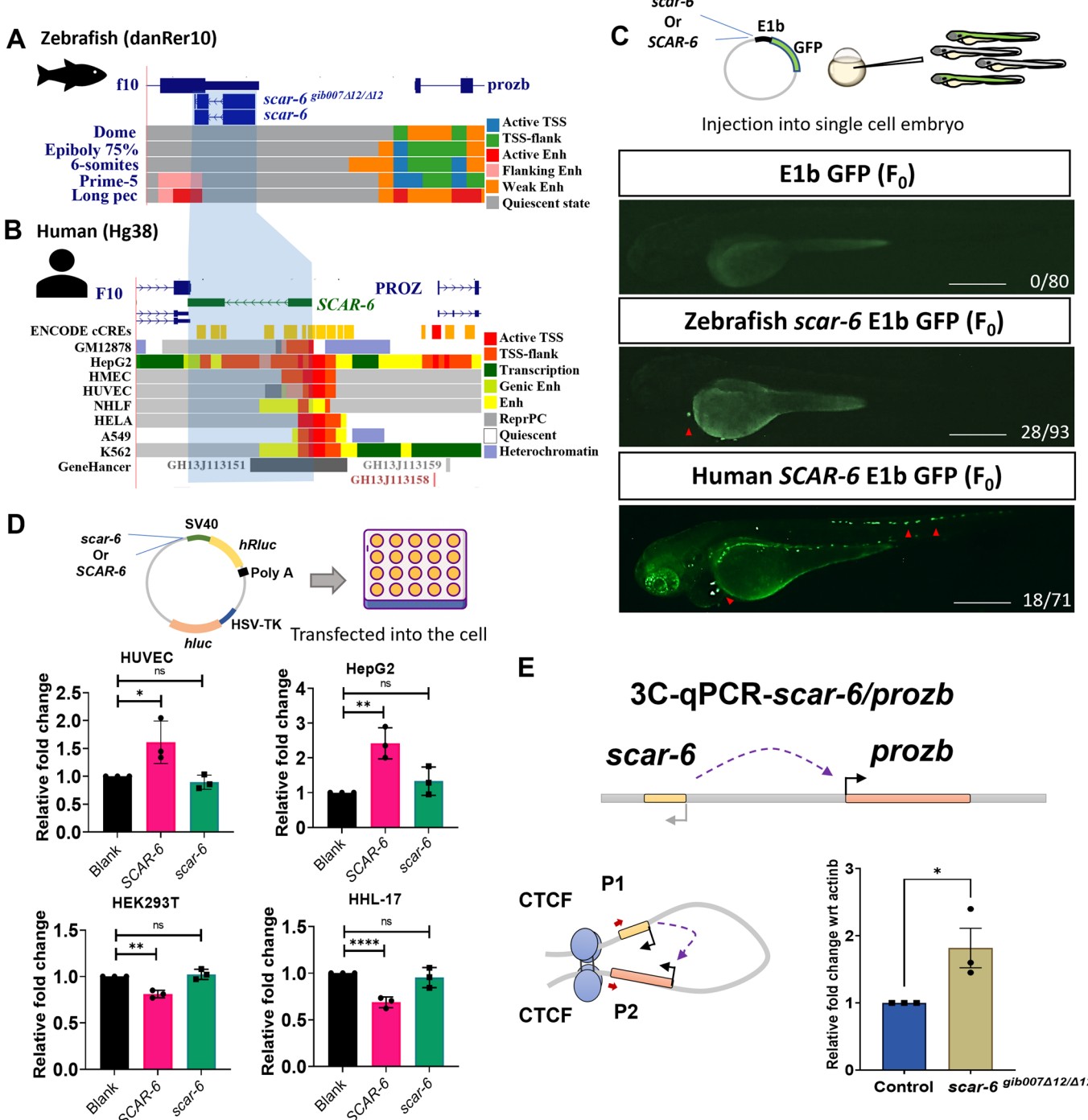

that the 12-bp deletion in *scar-6*$^{gib007\Delta12/\Delta12}$ animals encompassed the *prdm14* binding motif site and could possibly disrupt the binding of it (Fig. 5A). We further evaluated the *prdm14* binding at the *scar-6* locus using streptavidin-tagged DNA pulldown assay followed by immunoblotting using a specific *prdm14* antibody. Our experimental results revealed the presence of a distinct band, indicating the direct interaction between *prdm14* and the *scar-6* gene DNA locus. However, this band was notably absent in the *scar-6*$^{gib007\Delta12/\Delta12}$ gene DNA locus, indicating that the 12 bp deletion perturbed the binding of *prdm14* to

the *scar-6* locus (Fig. EV5D). We also performed ChIP-qPCR to assess the binding of *prdm14* protein at the *scar-6* locus in both wild-type and *scar-6*$^{gib007\Delta12/\Delta12}$ animals. We observed that *prdm14* binding was enriched in the wild-type zebrafish but didn't show any enrichment in the *scar-6*$^{gib007\Delta12/\Delta12}$ zebrafish (Fig. 5B). These findings together provide strong evidence that *prdm14* preferentially binds to the *scar-6* locus and is disrupted upon 12-bp deletion. As previous studies have shown that *prdm14* exhibits regulation of the target gene through methylation, we also investigated the methylation pattern of the nearby CpG

**Figure 4. The *scar-6* locus exhibits cis-regulatory enhancer signature.**

(A) UCSC genome browser snapshot of zebrafish *scar-6* locus with chromatin annotation marks from danio-code data across 5 developmental stages. (B) UCSC genome browser snapshot of human *SCAR-6* locus with regulatory signature track from ENCODE data, chromatin annotation of 8 different cell lines from Epigenome roadmap project and enhancer data from GeneHancer database. (C) Representative image of enhancer assay for human and zebrafish *scar-6* locus in zebrafish injected with empty E1b-GFP, *scar-6*-E1b-GFP and human *SCAR-6*-E1b-GFP. Transient expression was observed in the $F_0$ animals in the cardiac region and in case of human *SCAR-6* some expression was also observed in the trunk region. The red arrowhead indicates eGFP signal in the animal. 2.5× magnification; scale bar = 500 μm. (D) Bar plot of luciferase enhancer assay for human and zebrafish *scar-6* locus in HHL-17; HEK297T; HUVEC/hTERT; and HepG2 cell line transfected with zebrafish *scar-6*-psi-Chek2 and human *SCAR-6*-psi-Chek2 plasmid. Enhancer function was observed in HUVEC/hTERT; and HepG2 cell line. Data from 3 independent biological replicates plotted as mean fold change ± standard deviation; ns, not significant, *$p < 0.05$, **$p < 0.01$, ****$p < 0.0001$ (two-tailed unpaired t-test). (E) Representative image of 3C experimental for *scar-6* and *prozb* locus and bar plot representing relative fold change between wild type and *scar-6*$^{gib007\Delta12/\Delta12}$ mutant zebrafish normalized with actb. The *scar-6*$^{gib007\Delta12/\Delta12}$ zebrafish showed an increase in the *scar-6/prozb* looping. Data from 3 independent biological replicates plotted as mean fold change ± SEM. *$p < 0.05$ (two-tailed unpaired t-test). Source data are available online for this figure.

island using targeted bisulfite sequencing. Our analysis revealed a partial decrease in methylation levels from 100–95% in wild-type animals to 85–78% in *scar-6*$^{gib007\Delta12/\Delta12}$ animals (Fig. 5C). Upon further investigation, we observed that the CTCF binding at the *scar-6* locus also overlapped the CpG island. Hence, these data together suggest that the *scar-6* enhancer region is a sensitive region where the partial decrease in the CpG methylation caused by 12-bp deletion of the *prdm14* motif, leads to an increase in CTCF occupancy.

Previous investigations have demonstrated that elncRNAs originating from the regulatory loci have the ability to recruit or stabilize the binding of transcription factors at their specific loci. (Li et al, 2013; Hsieh et al, 2014; Pnueli et al, 2015; Ivaldi et al, 2018; Islam et al, 2023; Oksuz et al, 2023). Given the previous understanding of *PRDM14* as a repressor that recruits the *PRC2* complex to the target region, we investigated whether the *scar-6* elncRNA transcripts derived from the loci of *scar-6*, have any potential role in the recruitment or interaction with these proteins.(Payer et al, 2013; Yamaji et al, 2013). To explore this, we performed RNA immunoprecipitation (RIP) followed by qRT-PCR using antibodies against *prdm14* and *Suz12* (protein from *PRC2* complex) in zebrafish. Remarkably, we observed significant enrichment of *scar-6* transcript binding with both *prdm14* and *Suz12* proteins when compared to the IgG control. These findings suggest that *scar-6* elncRNA interacts with the *prdm14-PRC2* complex (Fig. 5D). Further to investigate into the role and interaction between *scar-6* lncRNA and *prdm14*, we utilized a morpholino to knockdown the lncRNA and assessed the binding efficiency of *prdm14* at the *scar-6* locus. The results revealed a notable ~50% reduction in *prdm14* binding following the lncRNA knockdown (Fig. 5E). These findings together suggest *scar-6* elncRNA interacts with the *prdm14-PRC2* complex and may potentially be involved in stabilizing these TF at its locus (Fig. 5F).

## PAR2-mediated NF-kβ signaling induces endothelial cell activation and vascular dysfunction

Coagulation cascade genes such as *FX* and thrombin have a serine protease domain that interacts with protease-activated receptors (PAR) to initiate endothelial cell activation causing an increase in the surface adhesion molecule. This leads to the initiation of atherosclerotic lesions and to vascular dysfunction in the organisms (Willis Fox and Preston, 2020; Heuberger et al, 2019; van den Eshof et al, 2017; Alberelli and De Candia, 2014). Structurally, *PROZ* also harbors the serine protease domain which is conserved across species and hypothetically can activate these PAR receptors (Sejima et al, 1990). To understand if the *prozb* can also activate endothelial

cells and cause hemorrhage, we overexpressed *prozb* in zebrafish. We observed hemorrhage in the cranial region of the zebrafish similar to the *scar-6*$^{gib007\Delta12/\Delta12}$ animals (Fig. 6A). We also found a proportion of embryos with trunk development defects (Fig. 6B). We next checked if endothelial cells are activated by looking at the levels of different PAR receptors (*PAR1, PAR2a, PAR2b, PAR3*) in our zebrafish mutants. We observed that the levels of *PAR2a* and *PAR2b* were 2-fold upregulated in the *scar-6*$^{gib007\Delta12/\Delta12}$ animals when compared to the wild type (Fig. 6C). Next, to check the cleavage of PAR2 in vivo, we performed a protease assay. We created a plasmid expressing a protein complex of GFP and Luciferase linked with the PAR2 cleavage domain (Byskov et al, 2020). We injected the IVT product of the GFP-PAR2-Luciferase (100 ng/μl) into the single-cell embryos of wild-type and incrossed *scar-6*$^{gib007\Delta12/+}$ zebrafish. We observed a increase in the protease activity of the PAR2 domain in the *scar-6*$^{gib007\Delta12/\Delta12}$ animals when compared to the wild-type which was in accordance with the increase in the *PAR2a* and *PAR2b* expression levels in the *scar-6*$^{gib007\Delta12/\Delta12}$ animals (Fig. 6D). This implies that upregulated levels of *prozb* in *scar-6*$^{gib007\Delta12/\Delta12}$ zebrafish activates the *PAR2* receptor. Next, we looked into the downstream pathway by which the activation of the *PAR2* receptor leads to vascular dysfunction. We looked into the levels of one of the downstream proteins NF-kβ, which has been previously reported to be upregulated upon *PAR2* activation (Heuberger et al, 2019; Sriwai et al, 2013; Minami et al, 2003, 2004). We observed that in our *scar-6*$^{gib007\Delta12/\Delta12}$ animals, the protein level of NF-kβ were upregulated compared to the wild type (Fig. 6E). Next, we looked into the expression levels of downstream target genes which are activated by NF-kβ and also associated with vascular functioning. We found the expression levels of *vcam2b* and *icam1* to be upregulated whereas the levels of iNO2a were downregulated in our *scar-6*$^{gib007\Delta12/\Delta12}$ animals (Fig. 6F). However, we did not observe any significant changes in other downstream target genes of NF-kβ associated with vascular functioning (Appendix Fig. S15). These findings are consistent with previous reports indicating that activation of the *PAR2* receptor can lead to the activation of NF-kβ, resulting in the upregulation of adhesion molecules such as *VCAM1* and *ICAM1* (Byskov et al, 2020; Sun et al, 2021; Minami et al, 2004; Bae and Rezaie, 2009; Buddenkotte et al, 2005). We hypothesise that the upregulated expression of these adhesion molecules may contribute to endothelial activation, leading to increased platelet adhesion to the endothelial cell surface (Bombeli et al, 1998). This pro-atherogenic effect could potentially explain the observed hemorrhage phenotype in *scar-6*$^{gib007\Delta12/\Delta12}$ animals (Fig. 6G) (Nakashima et al, 1998; Walpola et al, 1995; Heuberger and Schuepbach, 2019).

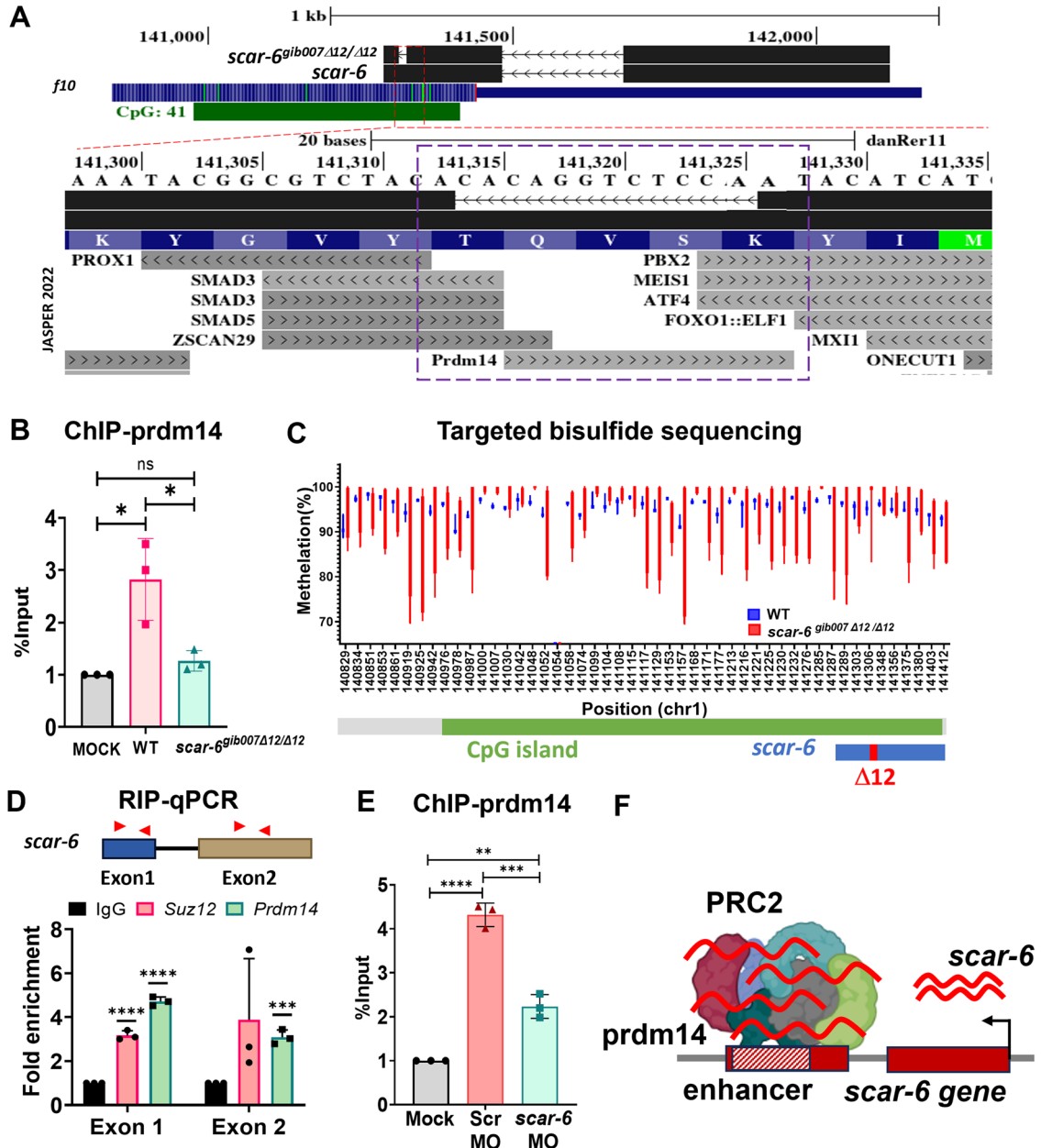

**Figure 5.  The *scar-6* elncRNA gene epigentically regulates expression of *prozb* via *prdm14*-PRC2 complex.**

(A) UCSC genome browser screenshot representing JASPER 2022 track with 12 bp deletion of *scar-6* locus. (B) Bar plot representing ChIP-qPCR quantification in percentage input using *prdm14* antibody as a target and mock as a control for *scar-6* locus in wild type and *scar-6^{gib007Δ12/Δ12}* mutant zebrafish. Data from 3 independent biological replicates plotted as mean fold change ± standard deviation; ns, not significant, *$p < 0.05$ (two-tailed unpaired t-test). (C) Box and whiskers plot representing percentage methylation profile of proximal CpG island of *scar-6* using targeted bisulfite sequencing of wild type and *scar-6^{gib007Δ12/Δ12}* mutant zebrafish per cytosine. Data from 4 independent biological replicates plotted with line at mean. (D) Bar plot representing fold enrichment of *scar-6* elncRNA upon RNA-immunoprecipitation of *prdm14* and *Suz12* followed by qPCR of *scar-6* elncRNA. The *scar-6* elncRNA shows enrichment in the binding with *prdm14* and *Suz12*. Data from 3 independent biological replicates plotted as mean fold change ± standard deviation; ***$p$-value ≤ 0.001, ****$p < 0.0001$ (two-tailed unpaired t-test). (E) Bar plot representing ChIP-qPCR quantification in percentage input using *prdm14* antibody as a target and mock as a control for *scar-6* locus in zebrafish injected with *scar-6* elncRNA morpholino (*scar-6* MO) and scrambled morpholino (scr MO). Data from 3 independent biological replicates plotted as mean fold change ± standard deviation; **$p < 0.01$, ***$p < 0.001$, ****$p < 0.0001$ (two-tailed unpaired t-test). (F) Schematic representation of *prdm14*-PCR-2 complex binding at *scar-6* locus and being stabalized by the *scar-6* elncRNA produced from the *scar-6* elncRNA gene. Source data are available online for this figure.

## Discussion

Poor sequence conservation in long noncoding RNA (lncRNA) necessitates exploring alternative modes of evolutionary conservation.

In this study, we investigated the functional role of a novel uncharacterized lncRNA locus called *scar-6* identified through a systematic approach in finding an lncRNA locus associated with cardiovascular function. Here, we report for the first time a lncRNA

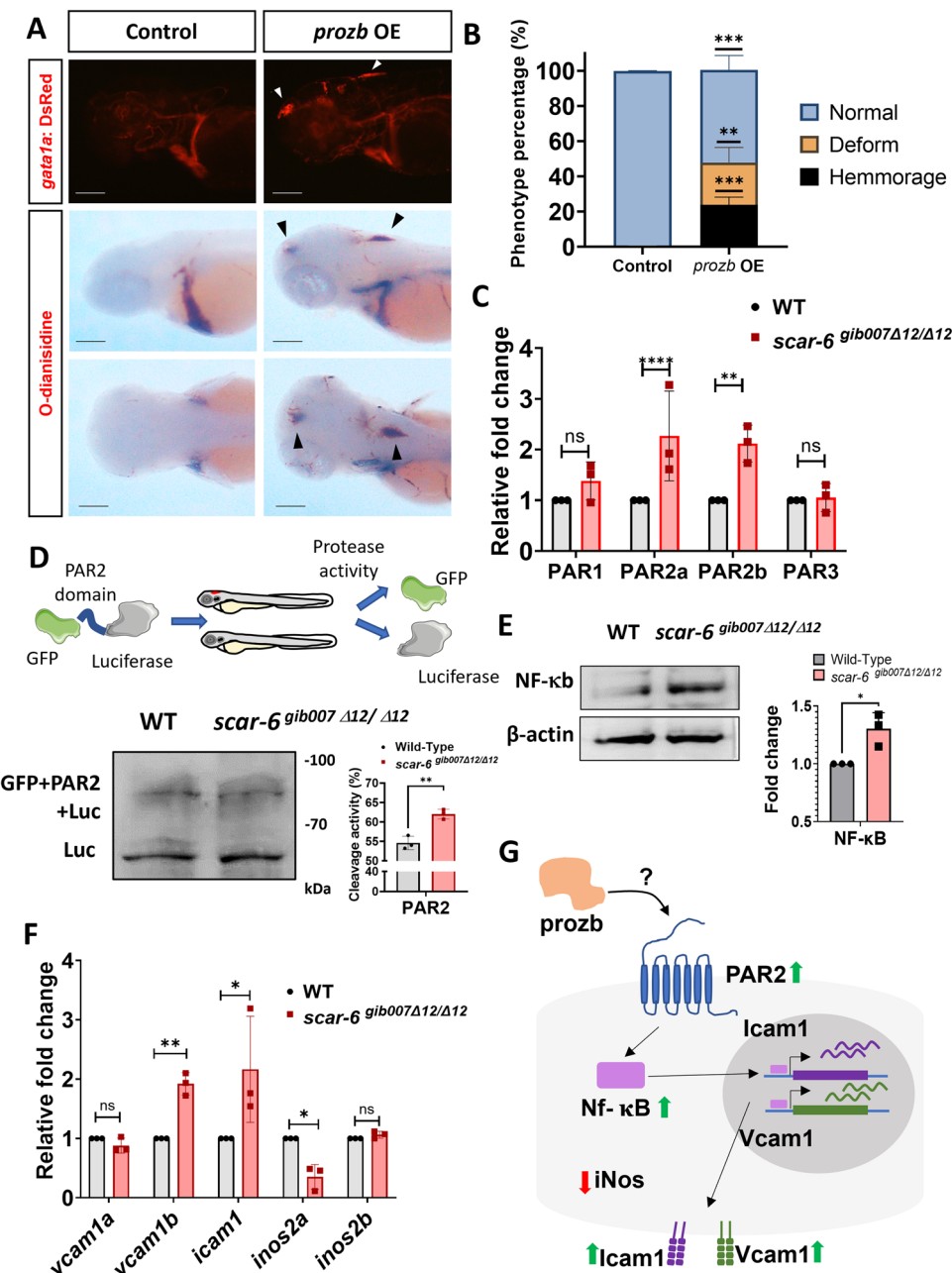

**Figure 6.** *Prozb* mediated endothelial cells activation via the PAR2-NF-kb pathway in *scar-6^{gib007Δ12/Δ12}* mutants.

(A) Representative image showing fluorescence for blood (*gata1a:DsRed*) in the cranial region of 3 dpf zebrafish and o-dianisidine stating of RBC blood cells in control and *prozb* IVT RNA (80 ng/μl) injected zebrafish. Black and white arrowhead denotes hemmorage in the animals. 4× magnification scale bar = 200 μm. (B) Stacked bar plot representing the phenotypic percentage of animals with haemorrhage and trunk deformities observed in control and *prozb* IVT RNA (80 ng/μl) injected zebrafish. Data from 3 independent biological replicates plotted as mean percentage ± standard deviation. **$p < 0.01$, ***$p < 0.001$ (unpaired two-tail t-test). (C) Bar plot representing relative fold change in expression of different PAR receptors (PAR1, PAR2a/b and PAR3) when compared between wild type and *scar-6^{gib007Δ12/Δ12}* mutant zebrafish. The PAR2a/b receptors expression levels were ~2 fold upregulated in *scar-6^{gib007Δ12/Δ12}* mutant zebrafish. Data from 3 independent biological replicates plotted as mean fold change ± standard deviation; ns, not significant, **$p < 0.01$, ****$p < 0.0001$ (two-tailed unpaired t-test). (D) Schematic representation of in-vivo PAR2 protease assay performed using eGFP-PAR2 domain-luciferase construct upon injecting into wild type and *scar-6^{gib007Δ12/Δ12}* mutant zebrafish followed by western blotting with luciferase antibody. The bar plot represents the quantification of cleavage activity in the wild type and *scar-6^{gib007Δ12/Δ12}* mutant zebrafish using the immunoblot quantification data. Data from 3 independent biological replicates plotted as mean ± standard deviation; **$p < 0.01$ (two-tailed unpaired t-test). (E) Western blot of NF-kB in wild type and *scar-6^{gib007Δ12/Δ12}* mutant zebrafish. The bar plot represents the quantification of the western blot from 3 independent biological replicates plotted as mean fold change ± standard deviation. *$p < 0.05$ (two-tailed unpaired t-test). (F) Bar plot representing relative fold change expression of *vcam1a, vcam1b, icam1, inos2a,* and *inos2b* when compared between wild type and *scar-6^{gib007Δ12/Δ12}* mutant zebrafish. Data from 3 independent biological replicates plotted as mean fold change ± standard deviation; ns, not significant, *$p < 0.05$, **$p < 0.01$ (two-tailed unpaired t-test). (G) Hypothetical schematic of *prozb* mediated endothelial cell activation through activation of *PAR2-NF-κB* pathway leading to upregulation of surface adhesion molecules (*Icam1* and *Vcam1*) and downregulation of *iNOS2a* causing vascular dysfunction and haemorrhage in zebrafish. Source data are available online for this figure.

locus that functionally modulates coagulation and vascular function in vivo. Interestingly, we observed that the RNA produced from the *scar-6* locus exhibits no sequence conservation with its putative human ortholog, although the locus itself is syntenic and functionally conserved. Dissecting the functional modality of the *scar-6* locus—whether as an RNA molecule, the act of transcription, or a DNA regulatory element—is intricate, as many lncRNA loci have demonstrated complex functions, exhibiting single or multiple functional modalities (Gil et al, 2023; Lewandowski et al, 2020; Andergassen et al, 2019; Santoro et al, 2013; Yin et al, 2015; Sehgal et al, 2021; Ritter et al, 2019; Wang et al, 2011). Functional assays, including knockdown, overexpression, locus editing, and complementation with the wild-type *scar-6* RNA copy in *scar-6*gib007Δ12/Δ12 animals, have elucidated that the *scar-6* locus encompasses a cell-type-specific distal enhancer, and also produces an RNA transcript that functions as an enhancer lncRNA. This elncRNA stabilizes the binding of transcription factors at the enhancer site on the *scar-6* locus. Although the *scar-6* RNA is involved in modulating in vivo hemostatic function, it did not show any independent impact on cranial hemorrhage. These findings underscore that the *scar-6* locus's function is mediated by both its DNA elements and the elncRNA it produces.

Moreover, the *scar-6* enhancer exhibits a regulatory effect on the *cis* protein-coding gene *prozb*. Bioinformatic analysis also suggests enrichment of DNA elements on the syntenic locus of conserved *scar-6* elncRNA between humans and zebrafish with sequence divergence (Ranjan et al, 2024). Previous functional studies have indicated that enhancer-associated syntenic lncRNAs exhibit low sequence conservation, implying limited evolutionary constraints on these noncoding regions (Hezroni et al, 2015; Gil et al, 2023; Paralkar et al, 2016; Rom et al, 2019). Furthermore, our detailed examination revealed that the *scar-6* gene locus partially overlaps with a conserved CpG island, with no sequence similarity. This CpG island in zebrafish is enriched with *prdm14* binding motifs, while in humans, it is enriched with *PRDM9* binding motifs that overlap with a tandem GC repeat region (Appendix Fig. S16). This raises a possibility that transposable elements-mediated rearrangements of the *scar-6* locus during evolution could have led to the gain or replacement of functional elements to adapt to the complex functioning of larger mammals (Modzelewski et al, 2022; Shapiro, 2014; Ciliberti et al, 2007; Ranjan et al, 2024). It might also provide a potential explanation for the enhancer signal observed in other regions of zebrafish injected with the plasmid containing the enhancer of the human *SCAR-6* locus. This type of vertebrate genome innovation has been previously reported between zebrafish and humans describing diversity in the function of noncoding regulatory regions. (Bedell et al, 2012; Lowe et al, 2011; Douglas and Hill, 2014; McEwen et al, 2009).

Multiple studies have demonstrated that RNA produced from the enhancer or promoter locus are called enhancer RNA(eRNA) or enhancer lncRNA based on the transcript characteristics and features. Briefly eRNA are transcript with single exon and bidirectional transcription and no poly A making it short lived. The other class of eRNA can be associated with the recruitment or stabilizing the binding of TF at the locus (Sigova et al, 2015; Andersson et al, 2014). Hence, based on RIP-qPCR data, the *scar-6* RNA could possibly recruit or stabilize the transcription factor *prdm14* and *PRC2* complex (*Suz12*) at the *scar-6* enhancer locus. Interestingly recent studies have highlighted that an organism exhibits certain regions in the genome that are sensitive to gene transcription regulation and are generally modulated by the

recruitment of a specific class of transcription factors called co-repressors at the enhancer locus to control the gene expression (Jacobs et al, 2023; Schertzer et al, 2019). In addition, the correlation between lncRNA potency, abundance, and PRC2 recruitment to CpG islands within lncRNA-targeted domains suggests a potential model. In this model, CpG islands autonomously attract PRC2 and interact with lncRNAs and their associated proteins in three-dimensional space, initiating the spread of PRC2 within these domains via lncRNA-independent mechanisms (Jacobs et al, 2023; Schertzer et al, 2019). In our study, we also observed a similar pattern whereupon disruption of the binding site of *prdm14-PCR2* complex (co-repressor) resulted in upregulation of the target gene *prozb*, leading to hemorrhage in the animals. This was in accordance with previous studies where the *prdm14-PCR2* complex has been associated with a repressor function where they modulate the expression of their target genes by methylating the CpG island nearby potentially involving Dnmt1 (Viré et al, 2006; Li et al, 2015). With these, we hypothesize that the *scar-6* locus is a sensitive region in the genome that modulates the expression of *prozb* gene by recruitment of the *prdm14-PRC2* complex as a co-repressor at its locus. The *prdm14-PRC2* complex represses the expression of *prozb* by methylating the nearby CpG island inhibiting the binding of CTCF to the locus and hampering the *scar-6* enhancer /*prozb* promoter looping (Fig. 7).

While only a few lncRNA locus have been associated with disease progression, a significant proportion of disease-related noncoding locus remain uncharacterized due to their complex manifestations, requiring the use of model organisms to decipher their functions (Feyder and Goff, 2016; Allou et al, 2021; Ishii et al, 2006; Vollmers et al, 2021; Sato et al, 2021). Interestingly, there have been no previous reports linking elncRNA locus to coagulation-related diseases, and investigations of coagulation-associated genes in mouse models have proven to be lethal (Jalbert et al, 1998; Ishiguro et al, 2000; Cui et al, 1996; Dewerchin et al, 2000; Weyand et al, 2019; Hu et al, 2017; Liu et al, 2014) However, zebrafish models have exhibited a certain degree of tolerance to knockout of coagulation genes, making them suitable for studying such genes (Liu et al, 2014; Hu et al, 2017; Weyand et al, 2019). In this study, we identified and characterized a novel elncRNA locus associated with hemostasis regulation. Precious editing of the elncRNA *scar-6* locus result in a hemorrhagic phenotype during early development, similar to the characteristic phenotype observed in coagulation-associated protein-coding genes (Hu et al, 2017; Weyand et al, 2019; Liu et al, 2014).

In the *scar-6* mutant, we did not observe any change in the expression of *f10* gene but found the *prozb* gene to be 8-fold upregulated. Prior research has emphasized the significant role of the *PROZ-ZPI* complex in regulating the activated *F10* proteins, which is a crucial gene in the pathway (Han et al, 2000, 1998; Sofi et al, 2004). Coagulation protein levels must be tightly controlled in a stoichiometric manner in normal physiological conditions. The *scar-6* lncRNA emerges as a critical and sensitive regulator of *prozb*, the primary inhibitor of *f10* protein. This highlights the crucial role of *scar-6* lncRNA in the coagulation pathway. Notably, we observed that the hemorrhage occurs early in *scar-6* locus mutants, This observation was similar to the late manifestation of hemorrhage in zebrafish knockouts for protein-coding coagulation genes such as *factor 10*, *factor V*, or *antithrombin III*, which typically occur during the juvenile or adult stages. Interestingly, we did not observe widespread hemorrhage similar to *f10*−/− mutants in *scar-6*gib007Δ12/Δ12 animals (Hu et al, 2017),

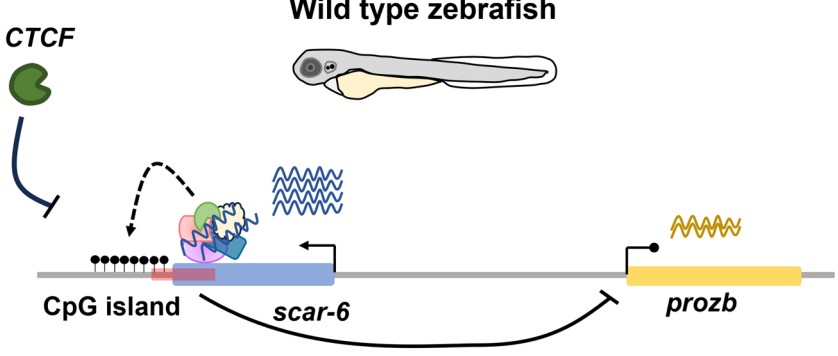

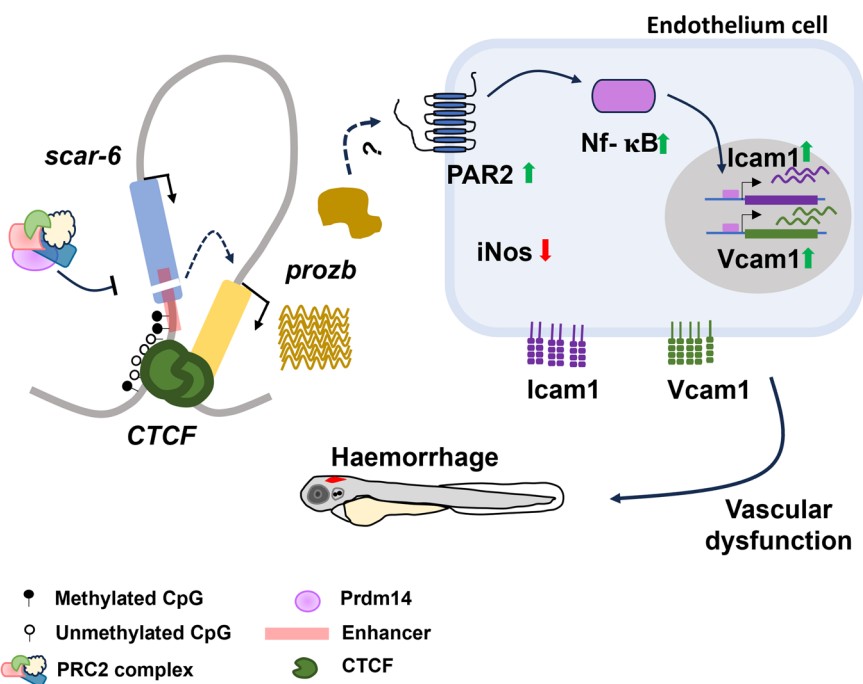

**Figure 7.  Epigenetic regulation of *prozb* through *scar-6* locus is important for vascular function and hemostatic process.**

Hypothetical schematic of *scar-6* locus in wild-type zebrafish, where *scar-6* elncRNA and *prdm14*-PRC2 complex hypermethylated the proximal CpG island and inhibits CTCF binding. In *scar-6^gib007Δ12/Δ12* mutant zebrafish. The 12 bp deletion affects the binding of the *scar-6* elncRNA and *prdm14*-PRC2 complex leading to a partial change in methylation. This change allows the CTCF occupancy at the locus further mediating sub-TAD looping of the enhancer-promoter of *prozb*. This causes upregulation of *prozb* leading to endothelial cell activation via the PAR2-NF-κB pathway and causing vascular dysfunction leading to hemorrhage.

and the expression level of *f10* remained unchanged. This suggests that the function of *f10* in the mutant was not altered. The molecular investigation provided insight that upregulation of *prozb* leads to activation of the *PAR2* receptor pathway causing vascular dysfunction and stroke-like phenotype. This finding aligns with previous reports in humans that have indicated altered levels of *Proz* in similar stroke-like phenotypes (Zhang et al, 2017; Staton et al, 2005). Although the exact mechanism of *Proz*-mediated PAR2 activation is still unknown and a detailed investigation is required to delineate the molecular interaction. Furthermore, the functional characterization of lncRNA locus provides valuable insights into the diverse manifestations of development and disease processes. Large or small structural variants such as the 13q34

deletion syndrome can manifest with a range of symptoms, including developmental delays, intellectual disabilities, facial and cranial abnormalities, stroke, heart defects, and various other physical and medical issues. The absence of genes in this specific region of chromosome 13 can profoundly influence both the physical and intellectual development of individual humans affected by this syndrome. Our identified lncRNA locus, *scar-6*, is also located within the 13q34 region and its functional study demonstrates its involvement in hemostasis and the maintenance of endothelial barrier function and its loss of function can lead to stroke-like phenotype as observed in the case of 13q34 deletion syndrome (Ponmani et al, 2015; Huang et al, 2012; Reinstein et al, 2016; Rath et al, 2015; Laurie and Bell, 2013).

These finding together highlights that the *scar-6* elncRNA locus is an important conserved regulatory locus in the organism that controls the expression of *prozb*, to maintain the normal levels of *coagulation factor Xa* in the organism and maintain proper homeostasis and vascular functioning in the organism.

## Methods

### Reagents and tools table

| Reagent/Resource | Reference or Source | Identifier or Catalog Number |
|---|---|---|
| **Experimental models** | | |
| *gib004Tg(fli1a*:EGFP; *gata1a*:DsRed) (*Danio rerio*) | Sridhar Sivasubbu lab, CSIR-IGIB (Lalwani et al, 2012; Lawson and Weinstein, 2002; Traver et al, 2003) | *gib004Tg(fli1a*:EGFP; *gata1a*:DsRed) |
| *Assam Wild Type zebrafish* | Sridhar Sivasubbu lab, CSIR-IGIB (Patoway et al, 2013) | *ASWT* |
| HUVEC-hTERT2 | ATCC | CRL-4053 |
| HepG2 | ATCC | HB-8065 |
| HLL-17 | Arvind H. Patel lab, University of Glasgow (Clayton et al, 2005)- a gift to Dr. Soumya Sinha Roy (CSIR-IGIB, India) | HHL-17 |
| HEK297T | ATCC | CRL-3216 |
| **Recombinant DNA** | | |
| Lenti-(BB)-EF1a-KRAB-dCas9-P2A-EGFP | Addgene | 118156 |
| E1b-GFP-Tol2 | Addgene | 37845 |
| TOPO pCR2.1-luciferase-PAR2-GFP | This study | |
| psi-CHEK-2 | Promega | C8021 |
| **Antibodies** | | |
| *CTCF* | Abcam | ab70303 |
| *IgG* | Abcam | ab231712 |
| *Suz12* | Cell Signaling Technology | ab3737 |
| *PRDM14* | Abcam | ab187881 |
| *NF-kβ p65* | Cell Signaling Technology | 3033 |
| *Firefly luciferase* | Thermo Scientific | PA5-32209 |
| *Factor X* | Abcam | ab97632 |
| *β-actin* | Cell Signaling Technology | 4967S |
| *Anti-rabbit IgG, HRP-linked Antibody* | Cell Signaling Technology | 7074S |
| **Oligonucleotides and other sequence-based reagents** | | |
| PCR primers | This Study | Eurofins |
| scar-6 Morpholino | This study | Gene tools |

| Reagent/Resource | Reference or Source | Identifier or Catalog Number |
|---|---|---|
| Scrambled MO | Gene tools | Vivo-Morpholino Standard Control Oligo |
| 5'biotin labeled scar-6 primer | This study | Eurofins |
| **Chemicals, Enzymes and other reagents** | | |
| Lipofectamine 3000 | Invitrogen | L3000015 |
| Dulbecco's modified eagle medium (DMEM) | Gibco | 10564011 |
| Fetal Bovine Serum | Sigma-Aldrich | F4135 |
| Vascular Cell Basal Medium | ATCC | PCS-100-030 |
| Endothelial Cell Growth Kit-BBE | ATCC | PCS-100-040 |
| Pierce™ Protein A/G magnetic beads | Thermo Scientific | 88803 |
| Dynabeads™ MyOne™ Streptavidin C1 | Thermo Scientific | 65001 |
| Dual-luciferase reporter assay kit | Promega | E1910 |
| Cas9 protein | Takara | 632641 |
| N- phenylthiourea | Sigma-Aldrich | P7629 |
| T7 mMESSAGE mMACHINE kit | Thermo Scientific | AM1344 |
| 3-aminobenzoic acid (Tricaine) | Sigma-Aldrich | E10521 |
| TB Green Premix Ex Taq II (Tli RNase H Plus) | Takara | RR82LR |
| BglII | NEB | R0144S |
| XhoI | NEB | R0146S |
| Kpn1 | NEB | R0142S |
| MseI | NEB | R0525S |
| Paraformaldehyde | Sigma-Aldrich | 158127 |
| DIG RNA Labeling Mix | Roche | 11277073910 |
| 1x TrypLE Express | Gibco | 12605010 |
| Collagenase IV | Gibco | 17104019 |
| Protease inhibitor cocktail | Roche | 04693159001 |
| Evans blue dye | Sigma-Aldrich | E2129 |
| Formaldehyde | Sigma-Aldrich | F8775 |
| RIPA lysis buffer | Thermo scientific | 89900 |
| Qiagen PCR purification kit | Qiagen | 28104 |
| SSRT II | Thermo scientific | 18064022 |
| T4 DNA ligase | NEB | M0202S |
| phenol:chloroform:isoamyl alcohol | Invitrogen | 15593031 |
| Epi-Tech Bisulfite conversion kit | Qiagen | 59104 |
| **Software** | | |
| Zfin | https://zfin.org/ | |

| Reagent/Resource | Reference or Source | Identifier or Catalog Number |
|---|---|---|
| STAR v2.7 | https://github.com/alexdobin/STAR | |
| Coding Potential Assessment Tool | http://lilab.research.bcm.edu/ | |
| Coding Potential Calculator | https://cpc.gao-lab.org/ | |
| DESeq2. | https://github.com/thelovelab/DESeq2 | |
| BS-Seq2 | https://github.com/HSiga/BSseq2 | |
| htseq-count | https://github.com/simon-anders/htseq | |
| GraphPad Prism v9 | Graph Pad | |
| RNAfold | http://rna.tbi.univie.ac.at/cgi-bin/RNAWebSuite/RNAfold.cgi | |
| JASPAR 2022 | https://jaspar.elixir.no/ | |
| Cas-OFFinder | http://www.rgenome.net/cas-offinder/ | |
| ZFLNC | https://www.biochen.org/zflnc/ | |
| ImageJ FIJI | https://github.com/fiji/fiji | |
| **Other** | | |
| MiSeq | Illumina | |
| Harvard Apparatus pump | Harvard Apparatus | |
| Stereo microscope Zeiss | ZEISS | |
| LightCycler LC480 | Roche | |
| CFX384 BioRad | BioRad | |
| Zeiss Axioscope A1 fluorescent microscope | Carl Zeiss | |
| Infinite® Lumi plate reader | TECAN | |
| Leica SP8 confocal microscope | Leica | |

## In silico selection of cardiovascular lncRNA and bioinformatics analysis

ZFLNC database was used in this study (Accessed Oct 1, 2018) (Hu et al, 2018). It contains a list of 13,604 lncRNA genes of zebrafish, obtained from RNA sequence data overlapped with Ensemble, NCBI, NONCODE, zflncRNApedia, and literature. All the lncRNA genomic coordinates were downloaded from the ZFLNC database. Using the ZFLNC database, we curetted out the list of lncRNAs that were conserved between zebrafish and humans. In the database, 3 modes of conservation were used to identify the conserved lncRNA. For our study, we selected the syntenic and collinearity model of conservation and retained those lncRNA transcripts. The selected sets of lncRNA genes were further analyzed for their genomic location, size, and adjacent 5 neighboring genes using the genome map of the ZFLNC database. We downloaded a list of

cardiovascular genes from Zfin using the terms heart, blood vessels, and endothelium cells. against the gene/transcript category individually (Bradford et al, 2022). The curated list of neighboring genes of lncRNAs was then overlapped individually with the list of cardiovascular genes. LncRNA with the overlapped neighboring genes were selected. Further, we prioritized the lncRNA genes having a single neighboring gene associated with any one of the selected tissues as they could be false-positive hits. In the end, we selected lncRNA which has more than 2 genes in synteny known to be functionally relevant in the cardiovascular system, their location, the importance of neighboring genes, disease association, and a number of human orthologs for further study.

We also computed the non-coding potential of the lncRNAs using bioinformatic algorithms such as the Coding Potential Assessment Tool (CPAT2) (Wang et al, 2013) or Coding Potential Calculator (CPC2) (Kang et al, 2017).

RNA-seq analysis was conducted as previously detailed (Ranjan et al, 2024) using publicly available datasets and reanalyzing them (Pauli et al, 2012; Yang et al, 2020; Sehgal et al, 2021). The reads were aligned to the zebrafish danRer11 reference genome using STAR v2.7. Raw counts were obtained with htseq-count, and normalization was performed using DESeq2. Expression data were plotted using GraphPad v9. Transcription factor binding at the *scar-6* locus was predicted using the JASPAR 2022 database (Castro-Mondragon et al, 2022) plugin track in the UCSC genome browser (Raney et al, 2024) for zebrafish danRer11 reference genome. In addition, the UCSC genome browser (Raney et al, 2024) track for the zebrafish danRer10 genome was utilized to visualize genomic locations, repeat regions, and epigenetic marks (Danio-code data; (Raney et al, 2024; Baranasic et al, 2022).

RNA seqcondary structure was predicted using RNAfold web server (https://rna.tbi.univie.ac.at/cgi-bin/RNAWebSuite/RNAfold.cgi) (Lorenz et al, 2011) by using the RNA sequence of *scar-6* lncRNA and 12-bp edited *scar-6* lncRNA with default setting.

## Animal study approval

The double transgenic zebrafish (*Danio rerio*) gib004Tg(*fli1a*:EGFP; *gata1a*:DsRed) (Lalwani et al, 2012; Lawson and Weinstein, 2002; Traver et al, 2003) and *scar-6*^gib007Δ12/+^ were maintained and handled in compliance with the protocols and guidelines approved by the Institutional Animal Ethics Committee of CSIR-Institute of Genomics and Integrative Biology, India (GENCODE-C-24). Special care was taken to ensure minimal distress to the animals.

## Microinjection of CRISPR-Cas9 and IVT products into the embryo

We made a cocktail of 25 ng/µl of sgRNA, 250 ng/µl of Cas9 protein (Takara, Japan), and 200 mM of KCl solution. The cocktail was preheated at 42 °C for 5 min and then incubated in ice. The cocktail was loaded into the needle using a microloader tip. The needle was fixed into the Injection holder (Narishige, Japan) and the tip was calibrated accordingly to inject 3 nl of the solution. Using the Harvard Apparatus pump (Harvard Apparatus, USA) 3 nl of the cocktail was injected into the single-cell stage embryos which were loaded on the embryo loading plate and visualized under a Stereo microscope Zeiss (ZEISS, Germany). Double transgenic zebrafish

gib004Tg(fli1: EGFP; gata1a: dsRed) embryos were used for injections (Lalwani et al, 2012; Lawson and Weinstein, 2002; Traver et al, 2003). After the injection, the embryos were incubated at 28.5 °C in system water for 8 hpf and then transferred to embryo water (E3) containing 200 mM (0.003%) PTU (N- phenylthiourea, Sigma Aldrich) and further incubated at 28 °C. The PTU embryo water was changed every 24 h and the dead and undeveloped embryos were removed simultaneously.

Similarly, scar-6, prozb and f10 were amplified from zebrafish cDNA and in-vitro transcribed using T7 mMESSAGE mMACHINE kit (Thermo Scientific), and different concentration was injected into the single cell stage of zebrafish as described above.

## Microinjection of morpholino and dCas9-KRAB into the embryo

Antisense splice blocking morpholino oligonucleotide (MO) was designed to target the 2nd exon of scar-6 (GeneTools, USA) (Appendix Table S4). Morpholino was dissolved in nucleus-free water to a final stock concentration of 1 mM. 3 nl of different concentrations of MO was injected into 1 cell stage of double transgenic zebrafish gib004Tg (fli1: EGFP; gata1a: dsRed) and the phenotype was observed. Scrambled MO (CCTCTTACCTCAGT-TACAATTTATA) was used for control injection.

Knockdown using dCas9-KRAB was performed as previously described (Long et al, 2015). We generated in vitro transcribed (IVT) RNA of the amplified dCas9-KRAB region from the Lenti-(BB)-EF1a-KRAB-dCas9-P2A-EGFP plasmid (a gift from Jorge Ferrer; Addgene plasmid #118156; http://n2t.net/addgene:118156; RRID) using the T7 mMessage mMachine kit. The sgRNA used in the knockout experiment was used. The injection was done at a final concentration of 300 ng/µl for dCas9-KRAB and 100 ng/µl for sgRNA. A single injection of dCas9-KRAB IVT was used as a control.

## Heteroduplex mobility shift assay (HMA)

We used heteroduplex mobility assay (HMA) to identify indels in the mutant animals as described previously (Sehgal et al, 2021; Ota et al, 2013). We amplified the 230 bp region around the target size with denaturing and slow cooling at the end. The amplicon products were run on a non-denaturing 15% DNA PAGE gel and stained using EtBr to observe the mobility shift in the heterozygous mutants. To identify homozygous mutants, we spiked a wild-type amplicon product in the PCR reaction followed by denaturing and slow cooling and the product was similarly run on non-denaturing 15% DNA PAGE gel.

## Mechanical vascular injury for coagulation assay

We performed mechanical injury in the posterior cardinal vein (PCV) as described previously (Clay and Coughlin, 2015). Briefly, we anesthetize the 3 dpf zebrafish embryos with 0.02% buffered 3-aminobenzoic acid (Tricaine). Using a minutia pin injury was made at the PVC of the zebrafish by piercing into it. The time for occlusion was recorded for up to 2 min (120 s) by a person who is blinded to the genotyping of the embryos. The individual embryos are genotyped after the assay is completed using HMA as described above.

## RT-qPCR

RNA isolation was carried out using 15 embryos with TRIzol (Invitrogen) and phenol-chloroform (Invitrogen). Subsequently, Turbo DNAse (Ambion) treatment and purification were conducted. cDNA was synthesized using 500 ng of total RNA and reverse transcriptase from Superscript II (Invitrogen). We used TB Green Premix Ex Taq II (Tli RNase H Plus) (TAKARA) to perform quantitative real-time PCR on LightCycler LC480 (Roche, Germany) or CFX384 BioRad. The relative levels of specific transcripts in the original pool of RNA were estimated using the methods described (Winer et al, 1999; Livak and Schmittgen, 2001). We used actb as a normalizing control for our analysis. The details for primers used for qRT-PCR are mentioned in the Appendix Table S4.

## Enhancer assay

An enhancer assay was performed using E1b-GFP-Tol2 (a gift from Nadav Ahituv; Addgene plasmid # 37845; http://n2t.net/addgene:37845; RRID:Addgene_37845). The target region was amplified and inserted into the plasmid using BglII and XhoI restriction enzyme (NEB) upstream to the E1B promoter. The construct was confirmed using target-specific PCR and Sanger sequencing. We injected the zebrafish scar-6 E1B-GFP and human SCAR-6 E1B-GFP plasmid into the wild-type ASWT zebrafish (75 ng/µl) (Patoway et al, 2013). The zebrafish was monitored till 5 dpf and positive animals were grown till adult and outgrossed to develop the stable lines. The imaged accusation was done on an upright Zeiss Axioscope A1 fluorescent microscope (Carl Zeiss, Germany).

The psi-CHEK-2 plasmid (Promega) was used for the luciferase assay. The target region was amplified and cloned upstream of the SV40 promoter using BglII and Kpn1 restriction sites. HHL-17, HEK297T, HUVEC/hTert2, and HepG2 were transfected with the cloned vectors using lipofectamine 300 (Thermofisher), and enhancer activity was calculated by normalizing renila luciferase signal with firefly luciferase signal.

## Whole-mount in situ hybridization (WISH)

With few modifications, WISH was performed according to a previously described protocol (Thisse and Thisse, 2008; Sehgal et al, 2021). Briefly, 3 or 5 dpf embryos were fixed in 4% paraformaldehyde (PFA) at 4 °C. The embryos were dehydrated and rehydrated using serial dilutions of methanol followed by proteinase K treatment. Hybridization of the DIG-UTP labeled RNA probe was performed overnight at 65 °C and staining was performed using NBT/BCIP (Sigma Aldrich) alkaline phosphatase substrates. Instead of PBST, we used tris-buffered saline and Tween 20 (1xTBST) buffer for stringent washing.

## Subcellular RNA isolation

The subcellular isolation of zebrafish was performed as previously described with few modifications (Ten et al, 2012). Briefly, 500 embryos of 3 dpf zebrafish were dissociated into single cells using 1x TrypLE Express (Gibco) and collagenase IV. The single cells were incubated and homogenized in hypotonic buffer (10 mM

HEPES, Ph 7.9; 1.5 mM MgCl$_2$; 10 mM KCl; 0.5 mM DTT; and Protease inhibitor) followed by centrifugation at 1000 rpm at 4 °C for 5 min. The supernatant was collated as cytoplasm and the pellet was washed with hypotonic buffer and collected as a nucleus. 1% of the sample was taken for western blotting and the rest was used for RNA isolation.

## Evans blue dye injection

Evans blue dye injection was performed as previously described with slight modifications (Smith et al, 2015). Briefly, 1% Evans blue dye was made in 1X Ringers solution and 5 nl/embryos was injected in the common cardinal vein (CCV). The injected embryos was incubated for 3 h. Confocal microscopy was performed for the dye at 620 ex/680 em. 10 regions in intersegmental vessels of control and mutant was selected and the mean gray value was quantified using ImageJ from 3 independent animals.

## ChIP-qPCR assay

The experiment was performed using the previously described method (Havis et al, 2006). Briefly, the pool of embryos (wt/scar-6$^{gib007\Delta12/\Delta12}$-20 embryos; scrMO/scar-6MO- 50 embryos) was fixed using formaldehyde and lysed using RIPA lysis buffer (Thermo Scientific). The sample was sonicated using a sonicator with 10 s on and 4 min off at 35% amplitude. The sample was centrifuged and incubated overnight in CTCF antibody (Abcam-ab70303) and was pulled using protein A/G beads (Thermo Scientific) and purified using Qiagen PCR purification kit and quantified by qPCR (primers in Appendix Table S4).

## RIP-qRTPCR

The experiment was performed using the previously described method (Sehgal et al, 2021; Li et al, 2014). Briefly, 500 embryos of 3 dpf zebrafish were lysed in RIPA buffer (Thermo scientific). The lysate was then UV crosslinked and then precleared in protein A/G beads. It was incubated with *IgG* (abcam-ab231712) *Suz12 (CST-ab3737)* and *prdm14 (abcam-ab187881)* antibodies overnight at 4 °C. The antibody was pulled using magnetic protein A/G beads and was treated in PNK buffer (10 mM NaCl, 10 mM Tri-Cl pH 7.6, 1 mM EDTA, 0.5% SDS) and then mixed in Trizol and the RNA was extracted out. An equal concentration of RNA was used to prepare cDNA using SSRT II (Thermo scientific) and using TB Green Premix (Takara) qRT-PCR was performed.

## Chromosome conformation capture (3C) assay

Chromosome Conformation Capture (3C) was performed as previously described with few modifications for zebrafish (Cope and Fraser, 2009). Briefly, the 20 embryos of 3 dpf zebrafish were fixed using 1X formaldehyde and lysed using a cell lysis buffer (10 mM Tris-Cl (pH 8.1), 10 mM NaCl, 0.5% NP-40, proteinase inhibitors). The sample was digested using MseI restriction enzyme (NEB) overnight at 37 °C, followed by ligation using T4 DNA ligase (NEB) overnight at 16 °C. The DNA was isolated using phenol:-chloroform:isoamyl alcohol (Invitrogen) method and the target region was quantified by qPCR (primers in Appendix Table S4).

## Targeted bisulfite sequencing

DNA was isolated for individual WT and scar-6$^{gib007\Delta12/\Delta12}$ zebrafish. Using Epi-Tech Bisulfite conversion kit (Qiagen) the DNA was treated following the instructions provided by the manual. The targeted region was amplified and proceeded with amplicon sequencing on MiSeq. BS-Seq was used to align the reads to the zebrafish (danrer11) genome and methylation percentage was calculated (Hansen et al, 2012).

## DNA pulldown assay

Target *scar-6* locus was amplified using 5'biotin labeled forward primer using WT and scar-6$^{gib007\Delta12/\Delta12}$ DNA as templates. The PCR product was purified and incubated with magnetic steptavidin beads for 1 h at room temperature then mixed with zebrafish lysate prepared from 500 embryos in RIPA lysis buffer (Thermo Scientific) and incubated overnight at 4 °C. The next day the magnetic beads were pulled down and dissolved in the SDS buffer and heated for 20 min at 95 °C and then proceeded with immunoblotting with the *prdm14* antibody.

## Statistical analysis

All the statistical analysis was performed using GraphPad Prism v9 or R v4.0.

## Western blotting

Western blot was performed as previously described. Briefly, the protein was isolated from zebrafish (NF-kβ p65—15 embryos; luciferase—1 embryo) using NP-40 lysis buffer (Thermo Scientific) with protease inhibitor was run on 10% SDS-PAGE gel and transferred to the PVDF or nitrocellulose membrane. The primary antibody of NF-kβ p65 (CST-3033) was diluted to 1:1000 and the blot was incubated overnight with the primary antibody. HRP conjugated secondary antibody was diluted 1:10,000 and incubated at room temperature and the blot was visualized using ECL under a chemiluminescence gel documentation system (Appendix Fig. S17).

## Cleavage assay

The firefly luciferase was amplified from psi-CHEK-2 plasmid (Promega), GFP was amplified from Lenti-(BB)-EF1a-KRAB-dCas9-P2A-EGFP plasmid (Addgene-118156) and PAR2 domain was amplified from HEK 293T cDNA and all three was cloned in frame together using In-Fusion cloning (Takara) in the background of TOPO pCR2.1 vector (Thermo Scientific) to generated luciferase-PAR2 domain-GFP vector. The IVT of luciferase-PAR2 domain-GFP was made using T7 mMassage mMachine kit (Thermo Scientific) and 100 ng/μl was injected into single cell stage zebrafish (WT and scar-6$^{gib007\Delta12/\Delta12}$). At 3 dpf the individual embryos were lysed using NP-40 lysis buffer (Thermo scientific) and western blotting was performed using Firefly luciferase (Thermo Scientific PA5-32209) at or 1:5000 dilution. HRP conjugated secondary antibody was diluted 1:10,000 and incubated at room temperature and the blot was visualized using ECL under a

chemiluminescence gel documentation system. Cleavage efficiency was calculated by the given formula.

$$\text{Cleavage efficacy}(\%) = [\text{luciferase intensity}/\text{Total (luciferase} \\ + \text{luciferase} - \text{PAR2} - \text{GPF})] \times 100$$

## Data availability

All the data has been provided in an Appendix file.

The source data of this paper are collected in the following database record: biostudies:S-SCDT-10_1038-S44319-024-00272-w.

## Peer review information

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

## Acknowledgements

Sridhar Sivasubbu like to acknowledge CSIR for the funding provided for this work (CSIR-MLP 2001). The authors would like to acknowledge Samatha Mathews for her input in designing experiments, discussions, and manuscript proofreading. We like to acknowledge Mercy Rophina and Arvind Kumar for their inputs in bioinformatics analysis. We acknowledge Hari Vignesh and Rahul Bhoyar for their help with sequencing. We also like to acknowledge Narendra Kumar for maintaining the zebrafish facility. We like to acknowledge Monika Verma, Dilip Kumar and Koushika Chandrasekaran for helping with the confocal microscope. We like to acknowledge reagent from Debojyoti Chakraborty (CSIR-IGIB) for CTCF antibodies. We would like to acknowledge Dr. Soumya Sinha Roy (CSIR-IGIB) for kindly providing the HHL-17 cells, originally a generous gift from Prof. Arvind H. Patel (MRC-University of Glasgow, UK).

## Author contributions

**Gyan Ranjan**: Conceptualization; Resources; Data curation; Software; Formal analysis; Validation; Investigation; Visualization; Methodology; Writing—original draft; Writing—review and editing. **Paras Sehgal**: Conceptualization; Validation; Methodology; Writing—review and editing. **Vinod Scaria**: Conceptualization; Software; Supervision; Validation; Writing—review and editing. **Sridhar Sivasubbu**: Conceptualization; Supervision; Funding acquisition; Validation; Methodology; Writing—review and editing.

Source data underlying figure panels in this paper may have individual authorship assigned. Where available, figure panel/source data authorship is listed in the following database record: biostudies:S-SCDT-10_1038-S44319-024-00272-w.

## Disclosure and competing interests statement

The authors declare no competing interests.

# Expanded View Figures

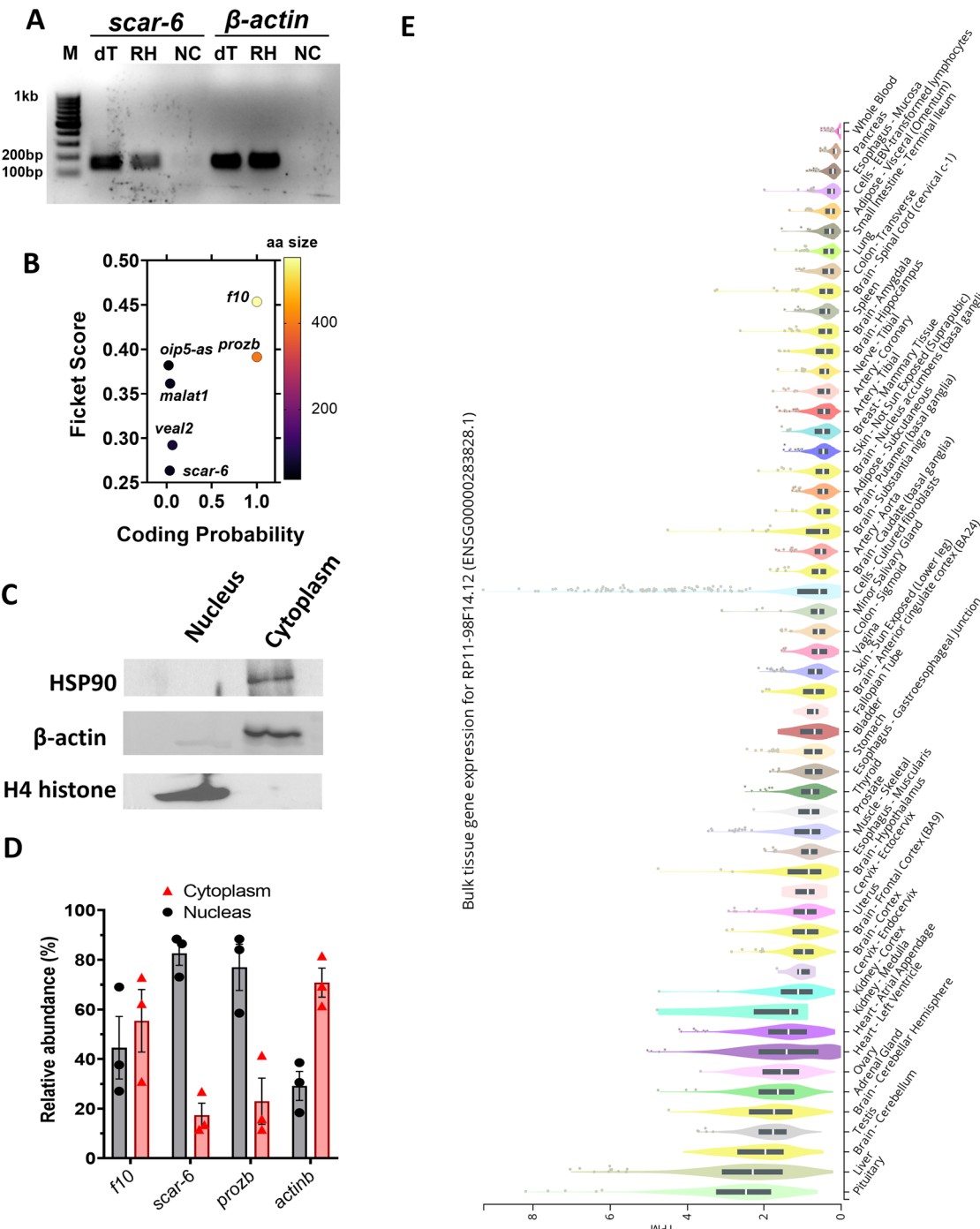

**Figure EV1.** *scar-6* lncRNA transcript is ubiquitously expressed and nuclear enriched.

(A) Image showing agarose gel electrophoresis of the PCR product derived from *scar-6* lncRNA and *β-actin*, amplified from cDNA synthesized using both oligo dT(dT) and random hexamer primers (RH). (B) Coding potentiality scores were calculated using CPC2 for *scar-6* and other lncRNA and protein-coding genes (*f10 and prozb*). (C) Western blot of *HSP90, β-actin* and H4 histone to confirm the purity of sub-cellular fractions of zebrafish cells. (D) The bar plot represents the relative abundance of the of *scar-6, f10* and *prozb* in different subcellular fractions quantified using qRT-PCR. The *scar-6* and *prozb* exhibit enrichment in the nucleus fraction and *f10* shows equal enrichment in cytoplasm and nucleus. Data from 3 different experiments were plotted as relative abundance percentages ± SEM. (E) The expression profile of human *SCAR-6* lncRNA across various tissues, as shown in the GTEx v8 database, is presented in transcripts per million (TPM). The box plot illustrates the data distribution, with the median indicated by a line inside each box, and the 25th and 75th percentiles represented by the lower and upper edges of the box, respectively. Outliers, defined as data points beyond 1.5 times the interquartile range, are also displayed. For sample sizes, please refer to the provided link: https://www.gtexportal.org/home/gene/ENSG00000283828.

## A  F₁ Generation of *scar-6* mutant (1M F₀ X WT) – HMA PAGE

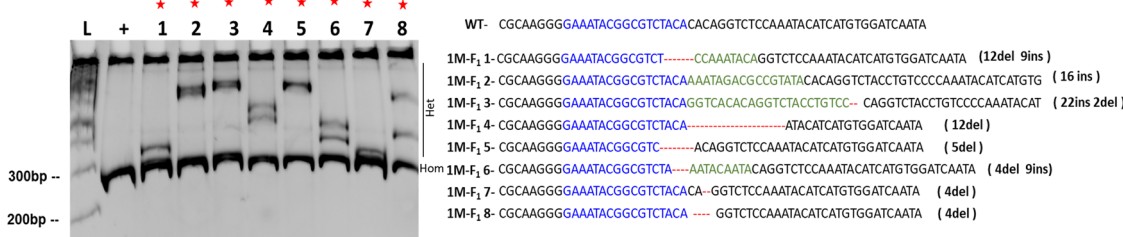

WT-    CGCAAGGGGAAATACGGCGTCTACACACAGGTCTCCAAATACATCATGTGGATCAATA

1M-F₁ 1- CGCAAGGGGAAATACGGCGTCT-------CCAAATACAGGTCTCCAAATACATCATGTGGATCAATA  ( 12del  9ins )

1M-F₁ 2- CGCAAGGGGAAATACGGCGTCTACAAAATAGACGCCGTATACACAGGTCTCACCTGTCCCCAAATACATCATGTG  ( 16 ins )

1M-F₁ 3- CGCAAGGGGAAATACGGCGTCTACAGGTCACACAGGTCTACCTGTCC-- CAGGTCTCACCTGTCCCCAAATACAT  ( 22ins 2del )

1M-F₁ 4- CGCAAGGGGAAATACGGCGTCTACA----------------------ATACATCATGTGGATCAATA  ( 12del )

1M-F₁ 5- CGCAAGGGGAAATACGGCGTC--------ACAGGTCTCCAAATACATCATGTGGATCAATA  ( 5del )

1M-F₁ 6- CGCAAGGGGAAATACGGCGTCTA----AATACAATACAGGTCTCCAAATACATCATGTGGATCAATA  ( 4del 9ins)

1M-F₁ 7- CGCAAGGGGAAATACGGCGTCTACACA--GGTCTCCAAATACATCATGTGGATCAATA  ( 4del )

1M-F₁ 8- CGCAAGGGGAAATACGGCGTCTACA ---- GGTCTCCAAATACATCATGTGGATCAATA  ( 4del )

## B

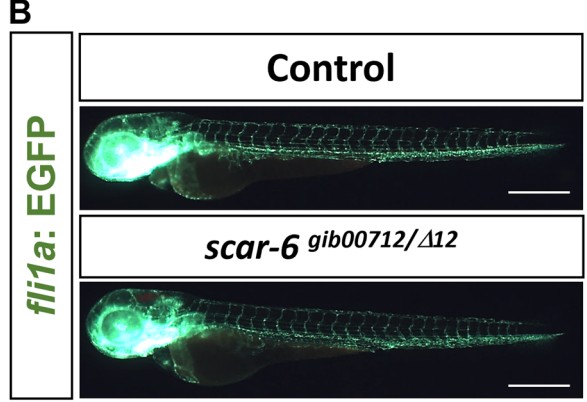

*fli1a*: EGFP

Control

*scar-6 ᵍⁱᵇ⁰⁰⁷¹²/Δ¹²*

## C

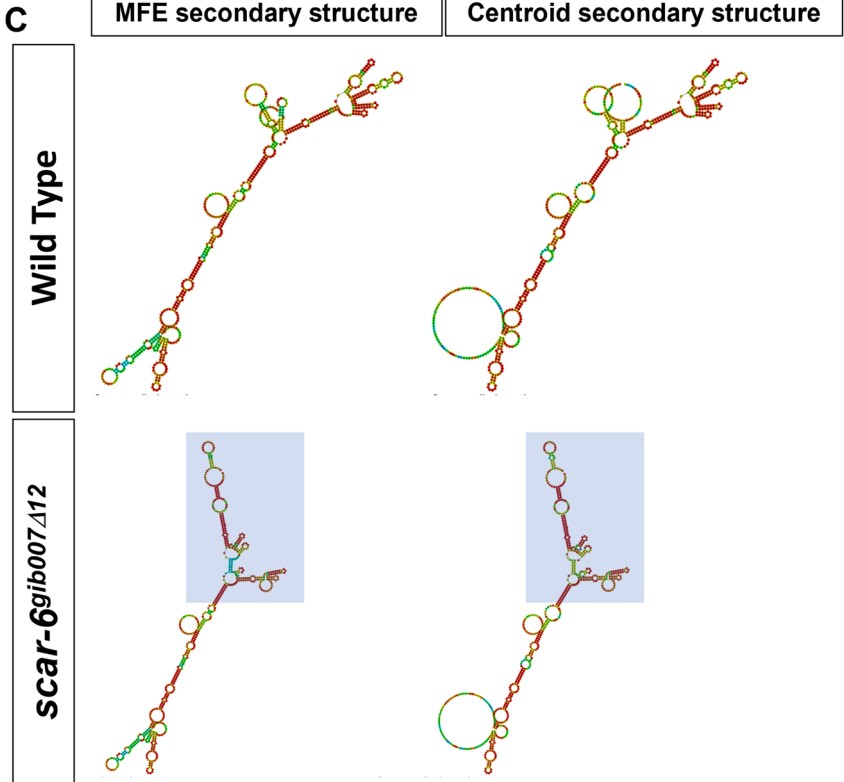

| MFE secondary structure | Centroid secondary structure |

Wild Type

*scar-6ᵍⁱᵇ⁰⁰⁷Δ¹²*

◀ **Figure EV2.  CRISPR-Cas9 mediated mutant generation of *scar-6* lncRNA gene.**

(A) DNA-PAGE gel illustrating the results of the heteroduplex mobility assay (HMA) conducted on the *scar-6* targeted region in F$_1$ *scar-6* mutant adult zebrafish, with genotypes identified by in-dels. Extra bands in the PAGE gel represent heteroduplex templates formed due to heterozygosity in the target region. The red asterisk denotes heterozygous mutant animals. (B) Representative image showing blood vessels of 3 dpf zebrafish progeny derived from wild-type gib004Tg(*fli1a*:EGFP;*gata1a*:DsRed) and *scar-6*^gib007Δ12/Δ12^ zebrafish. 2.5× magnification; scale bar = 500 μm. (C) Representative image showing secondary structure of scar-6 RNA under wildtype and 12-bp edited condition. The highlighted region represent change in the structure of lncRNA.

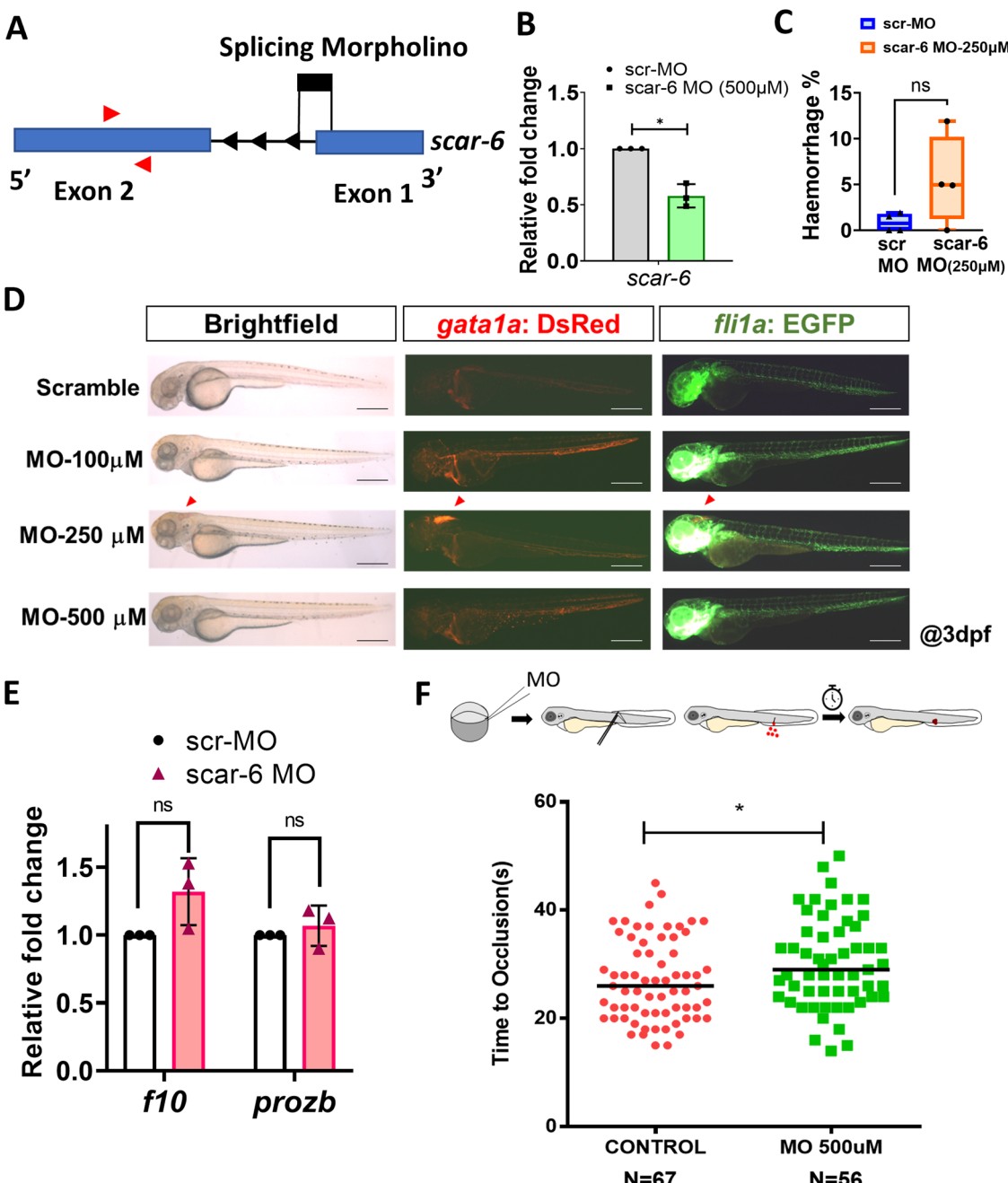

**Figure EV3. Knockdown of *scar-6* shows no significant phenotype.**

(A) Schematic of antisense splice blocking morpholino oligo targeting the exon 1 splicing junction of *scar-6* lncRNA. (B) Relative fold change expression of *scar-6* upon of knockdown with splice blocking morpholino in zebrafish at 3 dpf. Data from 3 independent biological replicates plotted as mean fold change ± standard deviation; *$P < 0.05$ (two-tailed unpaired t-test). (C) Box and whisker plot representing the percentage of animals with haemorrhage phenotype in zebrafish embryos injected with 250 μM of morpholino. Data from 4 independent experiments are represented by boxes indicating the interquartile range (25th to 75th percentiles), with the horizontal line within each box denoting the median. Whiskers extend to 1.5 times the interquartile range to define the minimum and maximum values, while individual points represent data from each experiment.: ns, not significant (two-tailed unpaired t-test). (D) Representative image of gib004Tg(*fli1a*:EGFP;*gata1a*:DsRed) zebrafish injected with different concentration of morpholino at 3 dpf. Red arrowhead denotes hemorrhage in the animals. 2.5× magnification, scale bar = 500 μm. (E) Relative fold change expression of *f10* and *prozb* upon knockdown of *scar-6* lncRNA with splice blocking morpholino in zebrafish. Data from 3 independent biological replicates plotted as mean fold change ± standard deviation; ns, not significant (two-tailed unpaired t-test). (F) Coagulation assay plot calculating the time of occlusion (s) in different individual zebrafish in control and 500uM morpholino injected zebrafish. Each point represents the occlusion time of individual zebrafish segregated based on genotype; *$P < 0.05$ (Mann-Whitney U).

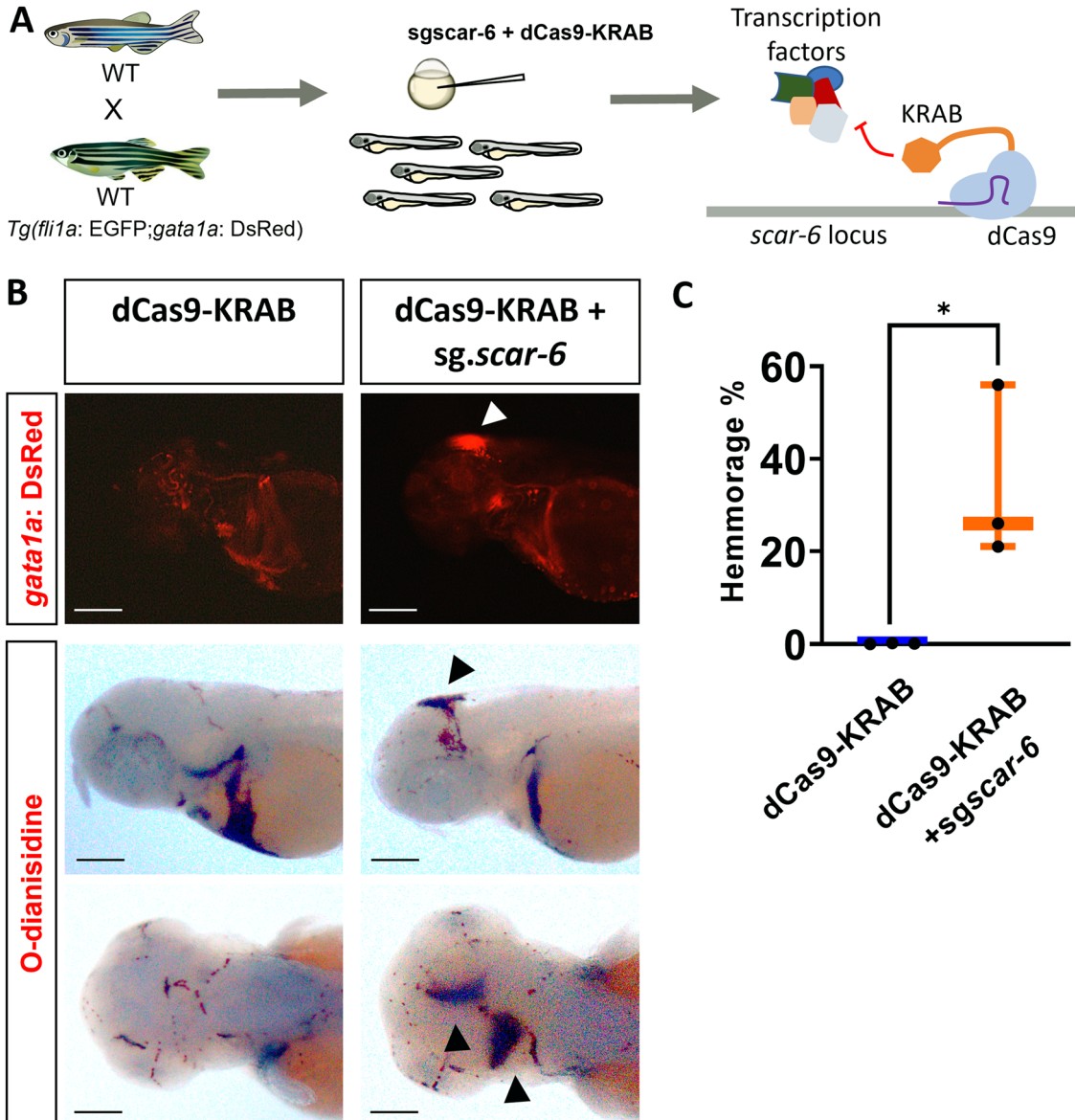

**Figure EV4. CRISPRi of *scar-6* locus exhibits haemorrhage phenotype.**

(A) Representation of CRISPR-dCas9-KRAB mediated inhibition of TF binding on *scar-6* locus. (B) Representative image showing fluorescence for blood (*gata1a:DsRed*) in the cranial region of 3 dpf zebrafish and o-dianisidine stating of RBC blood cells in control and dCas-9-KRAB + sg*scar-6* injected zebrafish. The black and white arrowhead denotes hemorrhage in the animals. 4× magnification; scale bar = 200 μm. (C) Box and whisker plot representing the percentage of animals exhibiting haemorrhage phenotype in control and dCas-9-KRAB + sg*scar-6* injected zebrafish. Data from 3 independent biological replicates are represented by boxes indicating the interquartile range (25th to 75th percentiles), with the horizontal line within each box denoting the median. Whiskers extend to 1.5 times the interquartile range to define the minimum and maximum values, while individual points represent data from each experiment; *$p < 0.05$ (two-tailed unpaired t-test).

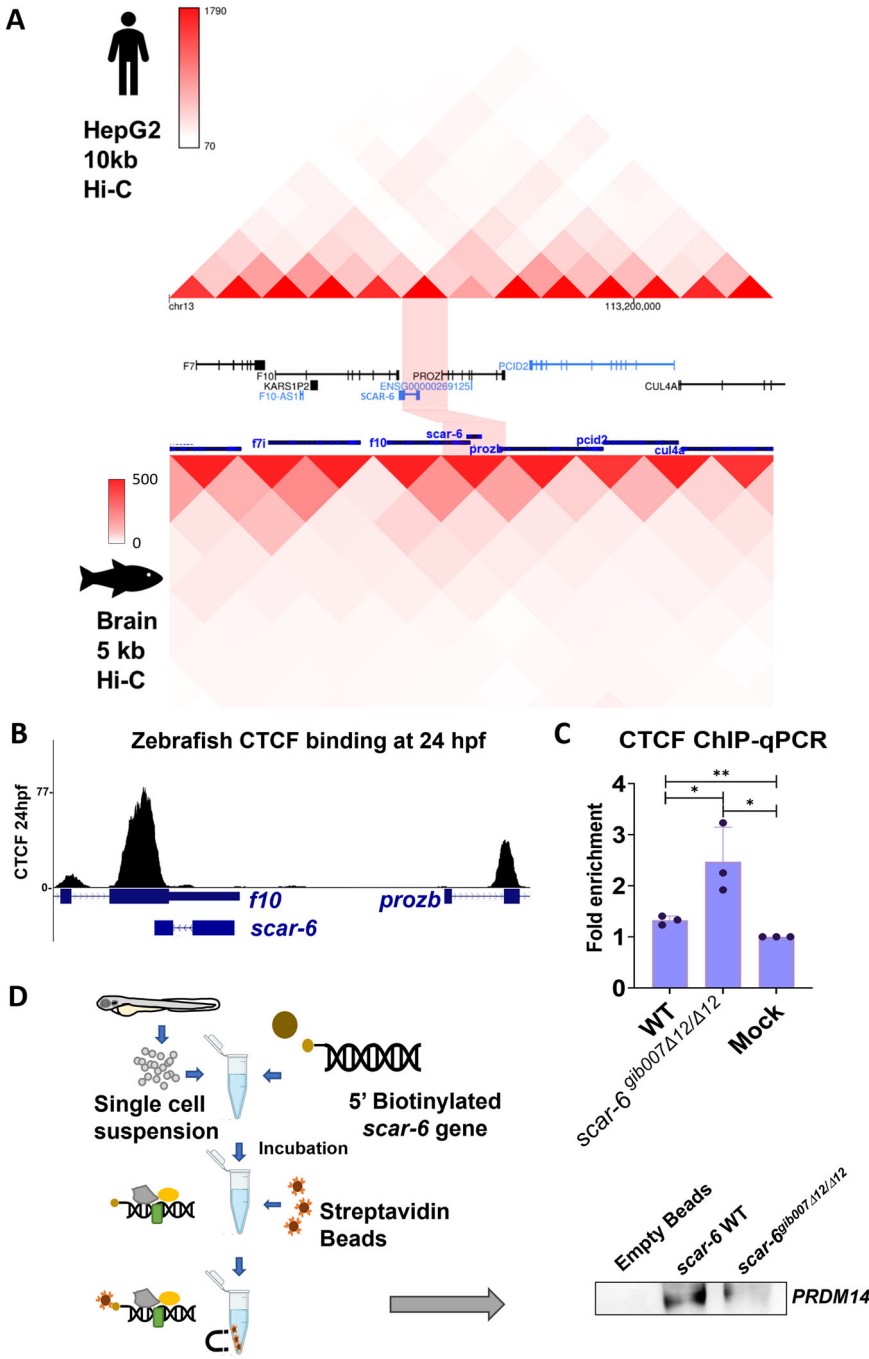

**Figure EV5.   Sub-TAD looping mediated enhancer-promoter interaction of *scar-6/prozb* locus.**

(**A**) Hi-C heatmap representation of Human *SCAR-6* locus in HepG2 at 10 kb resolution from ENCODE database (ENCODE Project Consortium et al, 2020; Wang et al, 2018) and zebrafish *scar-6* locus in brain tissue at 5 kb resolution from (Yang et al, 2020). (**B**) UCSC genome browser snapshot of data for CTCF binding peaks at 24 hpf of zebrafish for at *scar-6* and *prozb* locus (Pérez-Rico et al, 2020). (**C**) Bar plot representing ChIP-qPCR quantifying fold enrichment using CTCF antibody for *scar-6* locus in wild type and *scar-6^{gib007Δ12/Δ12}* mutant zebrafish. Data from 3 independent biological replicates plotted as mean fold enrichment ± standard deviation; *$P < 0.05$, **$P < 0.01$ (two-tailed unpaired t-test). (**D**) Schematic and western blot of DNA-pulldown assay was performed using streptavidin tagged scar-6 gene DNA in zebrafish, and immuno blotting was done using *prdm14* antibody. $N = 3$.

