## [Peer Review File · EMBO Reports]

SCAR-6 eIncRNA locus epigenetically regulates PROZ and modulates coagulation and vascular function

Gyan Ranjan, Paras Sehgal, Vinod Scaria, and Sridhar Sivasubbu

Corresponding author(s): Sridhar Sivasubbu (sridhar@igib.in), Vinod Scaria (vinod.scaria@vishwanathcancercare.org)

Review Timeline:

Submission Date:	4th May 24
Editorial Decision:	10th Jun 24
Appeal Received:	23rd Jul 24
Editorial Decision:	27th Aug 24
Revision Received:	3rd Sep 24
Accepted:	16th Sep 24

Editor: Esther Schnapp

Transaction Report:

Dear Dr. Sivasubbu,

Thank you for the submission of your manuscript to EMBO reports. We have now received the enclosed reports from 2 referees. Given that the 2 reports are in fair agreement, I am making a decision on your ms now in order to save you from unnecessary loss of time.

As you will see, none of the referees are very positive about the manuscript. Both point out that the current data are not sufficiently strong nor conclusive to support the main hypotheses. Referee 1 also rates the technical quality of the study "low" on the summary evaluation sheets returned with their reports.

Given these opinions and the fact that EMBO reports only invites revision of papers that receive enthusiastic support from the referees, I am afraid that we cannot offer to publish your manuscript.

I am sorry to disappoint you on this occasion, and hope that the referee comments will be helpful in your continued work in this area.

Yours sincerely

Referee #1:

The manuscript by Ranjan et al., describes the syntenic-based identification of the previously uncharacterized locus scar-6 in zebrafish that produces a long noncoding RNA transcript. The scar-6 locus is located in the vicinity of a probz gene which is known to regulate coagulation. Thus, the authors tested if the scar-6 locus has a role in coagulation and cardiovascular function by generating a mutant zebrafish that carries a very small deletion of 12-bp in the middle of the second exon of the scar-6 transcript. The authors reported coagulation deficits in animals carrying homozygous 12-bp deletion. As shown by several experiments, this coagulation function is not dependent on the lncRNA transcript produced from the locus but is regulated by an enhancer element embedded in the locus. This enhancer element regulates gene expression of the probz gene through binding of the transcription factor prdm14.

Whereas the described phenotype appears interesting and important, the premise of the study is flawed. The title of the manuscript, introduction, the first half of the results and discussion suggest that the cranial hemorrhage phenotype observed in the homozygous scar-6gib007Δ12 animals is due to the scar-6 lncRNA. However, multiple lines of evidence presented in the study demonstrate that the deficits are not triggered by the scar-6 lncRNA per se but are regulated by an enhancer element embedded in the locus. As such, the study requires substantial conceptual changes including changes in its title as there is not sufficient evidence for the role of the syntenically positioned lncRNA transcript (or its act of transcription). The only experiment suggesting a link to scar-6 RNA is RNA immunoprecipitation (RIP) showing enrichment of prdm14 and suz12 binding to the scar-6 transcripts. The two candidate proteins were chosen to be tested in RIP experiments based on the educated guess rather than an unbiased identification of scar-6 RNA protein interactors. Furthermore, the RIP experiment alone is not sufficient to support the model proposed in the manuscript. Together, the study finds an enhancer element that regulates cranial hemorrhage; there is no evidence for the functional importance or functional conservation of the syntenically positioned lncRNA scar-6. My specific concerns about the conceptual and technical quality of the study are summarized below.

1. Choice of the genome editing strategy to manipulate the scar-6 locus should be explained. The authors introduced a 12-bp deletion in the middle of the second exon of the scar-6 transcript. The rationale for this minimally invasive genome editing strategy should be provided. Dependent on the hypothesis (for example, that scar-6 acts through the act of transcription or as an RNA transcript), one could envision to target different regions of the scar-6 locus. Targeting exon-intron junctions with such a minimally invasive deletion could potentially lead to the destabilization of the whole scar-6. However, such a minimally invasive deletion in the middle of a non-conserved transcript requires thorough clarifications including analyses of its impact on the scar-6 transcript.
2. The authors should show if the removal of 12-bp in the second exon of scar-6 affects the expression levels of the scar-6 transcript at different time points. If the scar-6 levels remain unchanged in the homozygous deletion mutants, the authors should provide experimental evidence that indeed 12-nt removal in the scar-6 RNA causes observed phenotypes.
3. The logic and rationale for overexpressing scar6 RNA in zebrafish embryos is not clear. If elevation of scar6 lncRNA levels in their 12-nt deletion mutant was detected that would be reasonable to try to recapitulate the phenotype with the overexpression of the scar-6 transcript. In general, overexpression experiments are very difficult to interpret.

4. If scar-6 acts as a noncoding RNA transcript (at least partially), what does the missing 12-nt sequence within the mature lncRNA do? Why is it important for the lncRNA function?
5. Line 256 "We next evaluate the functional modality of the scar-6 lncRNA": rescue experiments using injections of in vitro transcribed scar-6 RNA would make sense only if scar6 lncRNA is degraded/downregulated in the 12-nt deletion mutant. As already mentioned above, the authors should test scar-6 expression in the mutant. Moreover, if the function of the scar-6 transcript indeed is dependent on the missing 12-nt, injections of mature scar-6 delta 12nt RNA should be used as a control for no rescue.
6. Line 273: "...the RNA does not appear to have any discernible impact or independent function on the cranial hemorrhage morphology or cis-regulatory effect on prozb". The sentence is misleading. As the authors inject mature (spliced, capped, polyadenylated) scar-6 RNA, cis-regulatory effect on prozb expression cannot be tested with this experiment. The cis regulatory effect of the scar-6 could have been tested if scar-6 was expressed from a DNA expression construct.
7. When overexpressing in vitro transcribed scar-6 RNA injected into the one-cell stage embryos, the authors measure time to occlusion at 3 days post fertilization (dpf). My concern is that the effect of RNA overexpression is scored at a very late developmental stage; it is unlikely that injected, in vitro transcribed RNA is stable until 3dpf. Same is valid for the following statement "We observed no prominent cardiovascular phenotype till 5dpf upon knockdown. This is a very late time point for a morpholino-induced phenotype to be evaluated.

Referee #2:

The manuscript by Gyan Ranjan et al. is about a previously uncharacterized lncRNA in zebrafish, which they named scar6. The lncRNA originates from a complex locus and partially overlaps with the gene f10 (Coagulation Factor X). The authors identified scar6 through syntenic analysis of lncRNA genes associated with the cardiovascular system. They describe that KO or deletion of 12nt of scar6 has severe physiological effects (mainly vascular; decreased survival, longer occlusion time, hemorrhage defects). Some effects could be rescued by injection of an in vitro transcribed version of scar6. Molecularly, the Delta12 scar6 had no significant effects on f10 or scar6 expression itself, however, it led to an upregulation of the neighboring gene prozb. The authors suggest that prdm14 binds scar6 RNA and genomic DNA, recruiting PRC2-subunit Suz12 to repress the expression of the lncRNA. This allows CTCF to induce prozb expression. Prozb activates the PAR2 system to induce in an NFkB-dependent manner the expression of cell adhesion molecules, which trigger cardiovascular dysfunction.

The manuscript is presented in a clear way, with minor language issues, and with a lot of data. The data clearly represents physiological importance of the molecules investigated. Also the molecular part was investigated. However, the highly complex locus of the lncRNA scar6 as well as the translation towards human cell lines/types remains difficult.

This is the first description of lncRNA scar6. I am not 100% convinced that this lncRNA really exists. Therefore, it is not clear, how long the RNA is and where does it exactly start and stop. Additionally, the lncRNA overlaps with F10, which itself has important functions for the cardiovascular system. It is therefore absolutely crucial to demonstrate that the effects seen are really from scar6 and not from a f10 mutant.

Major points:

1. As this is a novel lncRNA, several experiments are needed. The authors did some initial experiments, however, a full 5' and 3' RACE is needed incl Sanger sequencing. This would allow to see the start and end of the transcript. Additionally, it should be experimentally checked how long the transcript really is. For this, a northern blot would be greatly appreciated. Also key experiments should be performed with an exon-spanning primer. For me, when I blast the primer sequences provided, some bind also the overlapping gene f10, questioning whether the primers are really specific (especially 15/16/3/6).
2. It is not clear to me, why exactly these 12nt were selected for the mutation and not some other 12nt of scar6. Please explain.
3. Since the 12nt deletion of scar6 potentially overlaps with f10, it should be demonstrated that it is really the lncRNA mutation mediating this effect. This could be done by injecting an IVT of f10 (the overlapping part; so scar6 antisense) and testings for rescue.
4. Since the authors described a potential homolog of scar6 in humans, the relevance of this homologue should be tested. When I checked RNA-seq data in HUVEC, SCAR6 and the other genes weren't expressed, questioning the translational relevance of the findings. However, the authors should investigate in primary cells (not HUVEC/hTERT) from the cardiovascular system (e.g. HUVEC, smooth muscle cells, cardiomyocytes or similar) how highly SCAR6 is expressed and whether the knockdown also upregulate PROZB.
5. The authors showed that the 12nt deletion of scar6 led to an upregulation of prozb. What about the other genes in that locus (F7, PCID2, MCF2L)? Are they also affected? Is prozb the highest induced gene from the mutation - did the authors do a RNA-seq from the scar6 mutant?
6. The induction of prozb was only shown on RNA level. Please provide an evidence on protein level (western blot).
7. Why should prdm14 bind both, the RNA and the genomic DNA of scar6? As the genomic DNA binding and recruitment of PRC2 seems a logical explanation, the RNA binding of prdm14 seems contradictory as scar6 RNA would rather implicate that a

different function.

Minor points:

1. Introduction: Lines 61-65. I disagree with the statement that only a small subset of lncRNAs has been studied. The literature is enormous and since years every day new studies are published. Therefore i would recommend to reconsider this statement.
2. Introduction: Additionally, it should be mentioned that lncRNAs are physiologically important (see e.g. DOI: 10.1007/s00424-021-02641-z).
3. Introduction: Also the different ways how lncRNA can act should be mentioned (see e.g. DOI: 10.1038/s41467-024-45183-5)
4. Material and methods: 567-573: Which method was used to analyze RT-qPCR data?
5. Material and methods: The buffers used for several experiments are not listed in detail. What ingredients do they have? E.g. a hypotonic buffer was used for fractionation - this is not helpful and readers might have severe problems to validate their findings.
6. Maybe i did not find it, but the dCas System used is not explained in the methods.
7. The plasmids/vectors used in the manuscript are not explained in the material and methods. Please also indicate when they come from Addgene.
8. Please indicate also which software or program was used to analyze TFs, synteny, non-coding probability.
9. Which NGS tracks were used to show the genomic loci, their epigenetic environment and repetitive sequences in zebrafish? This must be mentioned in the material and methods. If e.g. UCSC was used, also this has to be cited.

** As a service to authors, EMBO Press provides authors with the ability to transfer a manuscript that one journal cannot offer to publish to another journal, without the author having to upload the manuscript data again. To transfer your manuscript to another EMBO Press journal using this service, please click on Link Not Available

Point to point response to reviewers comments.**Referee #1:**

The manuscript by Ranjan et al., describes the synteny-based identification of the previously uncharacterized locus *scar-6* in zebrafish that produces a long noncoding RNA transcript. The *scar-6* locus is located in the vicinity of a *prozb* gene which is known to regulate coagulation. Thus, the authors tested if the *scar-6* locus has a role in coagulation and cardiovascular function by generating a mutant zebrafish that carries a very small deletion of 12-bp in the middle of the second exon of the *scar-6* transcript. The authors reported coagulation deficits in animals carrying homozygous 12-bp deletion. As shown by several experiments, this coagulation function is not dependent on the lncRNA transcript produced from the locus but is regulated by an enhancer element embedded in the locus. This enhancer element regulates gene expression of the *prozb* gene through binding of the transcription factor *prdm14*.

Ans:- We sincerely appreciate the reviewer's diligent effort in comprehending our work in detail. The nomenclature of lncRNAs is often vaguely defined, leading to some confusion. We have reclassified and annotated our lncRNA as an enhancer lncRNA (elncRNA) based on its origin from an enhancer locus and its alignment with the current definition of enhancer lncRNAs.

Enhancer RNAs (eRNAs) are divided into two classes: (Lam MT, et al 2014;PMID: 24674738)

1. **Class 1 (2D-eRNA):** These eRNAs are bidirectional, single-exonic, non-polyadenylated, and short. Produced from enhancers, they are often considered transcriptional noise or features of enhancers.
2. **Class 2 (elncRNA/1D-eRNA):** These eRNAs exhibit characteristics similar to lncRNAs, such as unidirectional transcription, multi-exonic structure, lengths greater than 500 bp, and polyadenylation. elncRNAs show higher enhancer activity due to post-transcriptional processing and interact with transcription factors (TFs), stabilizing their binding at the locus and mediating transcription.

In brief enhancer lncRNAs (elncRNAs) are a subset of long non-coding RNAs (lncRNAs) that originate from enhancer regions of the genome. They often associated with regulation of gene expression by modulating the activity of their corresponding enhancers.

Our work provides evidence that *scar-6* elncRNA exhibits features of lncRNAs, including unidirectional transcription, a poly-A tail, a length greater than 500 bp, and a biexonic structure. It originates from an enhancer locus and helps stabilize the binding of the TF complex (*prdm14-PRC2*) to the enhancer on *scar-6* locus (Figure5E) .

Whereas the described phenotype appears interesting and important, the premise of the study is flawed. The title of the manuscript, introduction, the first half of the results and discussion suggest that the cranial hemorrhage phenotype observed in the homozygous *scar-6gib007Δ12* animals is due to the *scar-6* lncRNA. However, multiple lines of evidence presented in the study demonstrate that the deficits are not triggered by the *scar-6* lncRNA per se but are regulated by an enhancer element embedded in the locus. As such, the study requires substantial conceptual changes including changes in its title as there is not sufficient evidence for the role of the syntenically positioned lncRNA transcript (or its act

of transcription).

Ans- We express our sincere appreciation for the reviewer's diligent effort in comprehending the complexities of the lncRNA field. It is important to understand that the lncRNA field is complex and the lncRNA gene locus could have function beyond the RNA produced from its locus. (described in this reviews- Ali T, Grote P, 2020 ; PMID: 33095159; Kopp F, Mendell JT. 2018 PMID: 29373828; Mattick JS, et al 2023, PMID: 36596869).

It is common for the functionality of lncRNA genes to be misunderstood. Hence, we would like to highlight that the lncRNA genes locus can be functionally classified into 3 modes based on the functional modalities they possess (Kopp F, Mendell JT. 2018; PMID: 29373828)(These processes are widely accepted in the lncRNA field):

- The RNA produced from the gene is functional.
- The act of transcription of the gene is functional.
- The DNA elements present in the lncRNA gene body are functional. (Wu H, Yang L, Chen LL. 2017; PMID: 28629949.)

There are multiple lncRNA genes such as *AIRN* (Latos et al., 2012; Andergassen et al., 2019; Santoro et al., 2013), *Tug1*(Lewandowski, J.P. et al. 2020), and *Haunt* (Yin et al., 2015) which have more than one functional modality described above and can function through them.

In this manuscript, we focus on characterizing the entire *scar-6* locus, not just the transcript produced from its locus. This is why we have now modified the the title “*SCAR-6* elncRNA locus” to emphasize more on the locus. Our findings on the locus modality of the lncRNA locus as a DNA element are supported by multiple studies on candidate lncRNA gene locus studied by the experts in the field:

- *PVT1 lncRNA gene*- “Promoter of lncRNA Gene PVT1 Is a Tumor-Suppressor DNA Boundary Element”- Its promoter act as DNA boundary element (Cho SW et al, 2018 PMID: 29731168)
- *Lockd lncRNA gene*- Lockd gene positively regulates Cdkn1b transcription through an enhancer-like cis element (Paralkar VR, et al.,2016 PMID: 27041223)
- *Linc-p21 lncRNA gene* - “In Vivo Characterization of Linc-p21 Reveals Functional cis-Regulatory DNA Elements”- The cis-regulatory effects mediated by Linc-p21, in the presence or absence of transcription, are due to DNA enhancer elements. (Groff AF et al, 2016 PMID: 27524623)
- *Haunt lncRNA gene* - “Opposing Roles for the lncRNA Haunt and Its Genomic Locus in Regulating HOXA Gene Activation during Embryonic Stem Cell Differentiation”-The Haunt DNA locus contains potential enhancers of HOXA activation, whereas Haunt RNA acts to prevent aberrant HOXA expression. The lncRNA shows multifaceted function (Yin Y, et al 2015 PMID: 25891907)
- *Meteor lncRNAs locus* - “Complex regulation of Eomes levels mediated through distinct functional features of the Meteor long non-coding RNA locus” -distinct functional feature of the Meteor lncRNA locus-namely, the underlying DNA element-is required for suppressing Eomes expression following neuronal differentiation (Gil N, et al 2023, PMID: 37256750)

- *Tug1* lncRNA - “The *Tug1* lncRNA locus is essential for male fertility” - *Tug1* locus harbors two distinct noncoding regulatory activities, as a cis-DNA repressor that regulates neighboring genes and as a lncRNA that can regulate genes by a trans-based function (Lewandowski JP et al 2020 PMID: 32894169)

These lncRNA gene locus have shown a similar type of function through the DNA regulatory elements present on their locus.

Next we also observed that the lncRNA is originating from an enhancer locus and hence according to nomenclature definition *scar-6* is an enhancer associated lncRNA or elncRNA. We investigated the *scar-6* elncRNA role and found that it is involved in regulating the coagulation in zebrafish and also plays an important role in stabilizing the TF (*prdm14*) binding at the enhancer present on the *scar-6* locus. This type of function has previously been also reported for elncRNA /eRNA.

1. γ -globin/BGLT3 (Ivaldi MS et al 2018, PMID: 30150205),
2. E2/ER- α /eRNA (Li W, et al 2013 PMID: 23728302),
3. AR/CLK3 eRNA (Hsieh CL et al 2014 PMID: 24778216),
4. Cga/eRNA (Pnueli L, et al 2015 PMID: 25810254), and
5. INK4a/ARF (Farooq U, et al 2021 PMID: 33761351),

Our observation of the *scar-6* locus acting in a multifaceted function had also been previously reported from multiple other lncRNA locus. This is a common phenomenon of enhancer-associated lncRNA (elncRNA) gene that functions through the DNA element present on its locus. Here we present an investigation of a elncRNA locus which cis-regulated the *prozb* gene. The elncRNA locus is functional in two ways, first the enhancer element present of the locus performs enhancer function and the elncRNA transcript produced from *scar-6* locus helps in stabilizing the TF such as *prdm14* and *suz12* binding at the enhancer present on the *scar-6* locus.

As per the reviewers suggestion we have considered to change the title of the manuscript to “*SCAR-6* elncRNA locus epigenetically regulates *PROZ* and modulates coagulation and vascular function” which might be appropriate to the current work.

The only experiment suggesting a link to *scar-6* RNA is RNA immunoprecipitation (RIP) showing enrichment of *prdm14* and *suz12* binding to the *scar-6* transcripts. The two candidate proteins were chosen to be tested in RIP experiments based on the educated guess rather than an unbiased identification of *scar-6* RNA protein interactors. Furthermore, the RIP experiment alone is not sufficient to support the model proposed in the manuscript. Together, the study finds an enhancer element that regulates cranial hemorrhage; there is no evidence for the functional importance or functional conservation of the syntenically positioned lncRNA *scar-6*. My specific concerns about the conceptual and technical quality of the study are summarized below.

Ans: We appreciate the reviewers' feedback regarding the need for unbiased identification of lncRNA-protein interactions. This approach is essential when we lack prior knowledge of potential interactors or are conducting exploratory investigations. However, in our study, we specifically identified that the precise editing of the *scar-6* locus has disrupted the binding motif of the transcription factor *prdm14* (Figure 5A). Our objective was to determine if the elncRNA originating from the same locus could also interact with *prdm14*. Previous studies have indicated that RNA binding to TF can enhance their function by facilitating dynamic interactions between DNA, RNA, and TFs on chromatin (Oksuz O, et al., 2023, PMID: 37402367). We observed a significant reduction in *prdm14* binding at *scar-6* locus upon knockdown of the elncRNA *scar-6* using morpholino (Figure 5E). Furthermore, *prdm14* has been reported previously to bind to the PRC2 complex and is implicated in epigenetic regulation at the locus (Chan YS, et al 2013 PMID: 23280602; Yamaji M, et al 2013 PMID: 23333148; Nakaki F, Saitou M. 2014 PMID: 24811060; Payer B, et al 2013 PMID: 24268575). Hence we selected these 2 proteins to test our hypothesis.

We have also provided multiple other experiments which suggest that the elncRNA produced from the locus has a function. These are

- Overexpression of *scar-6* elncRNA enhances coagulation time. (Figure 3B)
- Knockdown using morpholino affects coagulation. (Figure EV3F)
- Rescue of coagulation defect upon supplementation with WT *scar-6* RNA and not with 12bp edited *scar-6* RNA. (Figure 3F)
- RNA-Immunoprecipitation of *scar-6* elncRNA with *prdm14* and *suz12* (Figure 5D)
- Knockdown of *scar-6* elncRNA using morpholino affected the binding of *prdm14* at the *scar-6* locus. (Figure 5E)

Our manuscript provides evidence that the *scar-6* locus is conserved between humans and zebrafish, and the functional significance of the enhancer element is also conserved. Both human and zebrafish loci produce bi-exonic elncRNAs, although these do not exhibit sequence conservation. We acknowledge the reviewers' comments and agree that further focused investigation is required to fully understand the conservation of the lncRNA, an area which remains under active investigation and is not comprehensively addressed in the current study. We have revised the manuscript to emphasize that the observed effects are due to the *scar-6* locus, while also suggesting the potential involvement of the elncRNA.

1. Choice of the genome editing strategy to manipulate the *scar-6* locus should be explained. The authors introduced a 12-bp deletion in the middle of the second exon of the *scar-6* transcript. The rationale for this minimally invasive genome editing strategy should be provided. Dependent on the hypothesis (for example, that *scar-6* acts through the act of transcription or as an RNA transcript), one could envision to target different regions of the *scar-6* locus. Targeting exon-intron junctions with such a minimally invasive deletion could potentially lead to the destabilization of the whole *scar-6*. However, such a minimally

invasive deletion in the middle of a non-conserved transcript requires thorough clarifications including analyses of its impact on the scar-6 transcript.

Ans: In our lab, we have generated multiple lncRNA knockouts in zebrafish using minimally invasive precise genome editing techniques. We have observed prominent phenotypes resulting from small indels. One of our study, published in The EMBO Journal, demonstrated that an 8 bp deletion can significantly disrupt *VEAL2* lncRNA function in zebrafish (Sehgal P, et al 2021, PMID: 34180064;). Additionally, several studies have shown that specific motif deletions, such as those in cyrano, can result in observable phenotypes (Goudarzi et al., 2019; PMID: 30620332). Based on this and our previous experience, we used a strategy to target both exons of the lncRNA. We anticipate that indel mutations will disrupt the secondary or functional structure of the lncRNA, rendering it non-functional. We also used F₀ screening strategy to select our mutant. Our previous observations, supported by multiple studies, indicate that CRISPR-Cas9 mediated editing and phenotypic screening in zebrafish can be efficiently conducted at the F₀ generation (Kroll F, et al 2021, PMID: 33416493; Lu F et al 2023 PMID: 36823429; Deniz E, et al 2018 PMID: 30151766; Parvez S et al 2023 PMID: 37069311a). This approach has proven to be an effective strategy for selecting mutants, thereby reducing both time and resources required for subsequent generations.

In the current study, we systematically designed two sgRNAs for our target *scar-6* lncRNA gene. One sgRNA targeted the 5' region (exon1) of the lncRNA gene, but F₀ screening of these animals injected with CRISPR-Cas9 RNP complex resulted in no substantial phenotypic changes. Conversely, the sgRNA targeting the 3' region (exon2) of the lncRNA gene exhibited a pronounced hemorrhage phenotype during our initial F₀ screening, prompting us to pursue this lead. We did had other mutants with different indels and 12 bp was the biggest deletion we got from our screening. We followed the 12 bp deleted animals for our study, emphasizing the specific and pivotal role of the 12bp deletion in influencing the observed phenotype and underscoring its functional importance in the context of our investigation. This is represented below and is given in the revised manuscript Supplementary Figure 6.

Supplementary Figure 6: - Representative image showing transgenic gib004Tg(*fli1a*:EGFP;*gata1a*:DsRed) 3dpf zebrafish injected with RNP complex of CRISPR-Cas9 targeting 5' and 3' region of *scar-6* gene (5x resolution)

2. The authors should show if the removal of 12-bp in the second exon of scar-6 affects the

expression levels of the *scar-6* transcript at different time points. If the *scar-6* levels remain unchanged in the homozygous deletion mutants, the authors should provide experimental evidence that indeed 12-nt removal in the *scar-6* RNA causes observed phenotypes.

Ans: We performed qRT-PCR to check the expression of the *scar-6* elncRNA transcript at 24 hpf and 72 hpf in the *scar-6*^{gib007Δ12/Δ12} zebrafish and observed that there was no significant change in the *scar-6* RNA levels (figure 2J) (3dpf) (24 hpf not show in the manuscript.) We suspect that the 12 bp deletion would have disrupted the functional structure of the RNA which is depicted in Fig EV 2C.

Line 256-258:

“However, we did not find any significant changes in the expression levels of *f10* and *scar-6* when comparing the *scar-6*^{gib007Δ12/Δ12} animals to the wild-type counterparts. (Fig 2J).”

Figure 2 [J] Bar plot representing relative fold change in expression levels of *f10*, *scar-6* and *prozb* between wild-type and *scar-6*^{gib007Δ12/Δ12} zebrafish. The *prozb* expression levels were 8-fold increase in the *scar-6*^{gib007Δ12/Δ12} zebrafish compared to the wild-type. Data from 3 independent biological replicates plotted as mean fold change ± standard deviation; *** p-value ≤ 0.001 (unpaired two-tail t-test)

In our manuscript, we provide evidence that a 12 bp deletion is indeed responsible for the coagulation phenotype. We demonstrated that the coagulation defect in the mutant could be rescued by introducing a wild-type (WT) copy of the *scar-6* elncRNA. Additionally, we performed experimental attempt to rescue the mutant with the 12-bp region deleted elncRNA, which did not result in any significant changes in the coagulation phenotype. This strongly indicates that the *scar-6* elncRNA regulates coagulation in zebrafish, and that the 12 bp deletion adversely affects its expression and function. (Figure 3F)

Figure 3 [F] Coagulation assay plot calculating the time of occlusion (s) in different individual zebrafish in control, in-cross progeny of $scar-6^{gib007\Delta12/+}$, in-cross progeny of $scar-6^{gib007\Delta12/+}$ injected with WT $scar-6$ IVT RNA (100ng/ul) and in-cross progeny of $scar-6^{gib007\Delta12/+}$ injected with $scar-6^{gib007\Delta12}$ IVT RNA (100ng/ul) performed blinded. Each point representing the occlusion time of individual zebrafish segregated based on injected concentration; ns- not significant ** $p < 0.01$ **** $p < 0.0001$ (Mann-Whitney U)

3. The logic and rationale for overexpressing scar6 RNA in zebrafish embryos is not clear. If elevation of scar6 lncRNA levels in their 12-nt deletion mutant was detected that would be reasonable to try to recapitulate the phenotype with the overexpression of the scar-6 transcript. In general, overexpression experiments are very difficult to interpret.

Ans: The rationale for performing overexpression of $scar-6$ lncRNA was to understand the function of the lncRNA in an overexpressed state. Given that we observed coagulation defects in the $scar-6$ mutant, we aimed to determine whether overexpression of the lncRNA could have an opposing effect or enhance coagulation in vivo. We acknowledge the reviewer's point that overexpression studies can sometimes be challenging to interpret; however, in our case, the coagulation assay provides a clean readout. Moreover, multiple studies have previously demonstrated the function of lncRNAs or circRNAs in zebrafish through overexpression. For example, overexpression studies show functions of lncRNA *VEAL2* (Sehgal P, et al 2021, PMID: 34180064;), lncRNA *durga* (Sarangdhar MA, et al 2017, PMID: 28442991), lncRNA *lnc304* and *lnc172* (Li W et al 2023, PMID: 36831339), and *CDR1as* (Memczak S, et al 2013 PMID: 23446348.) in zebrafish.

4. If scar-6 acts as a noncoding RNA transcript (at least partially), what does the missing 12-nt sequence within the mature lncRNA do? Why is it important for the lncRNA function?

Ans: The structure of lncRNA plays an important role in its function. Hence we examined the secondary structure of the wild-type and 12 bp edited *scar-6* lncRNA transcripts using RNAfold web server. Our analysis revealed a significant alteration in the secondary structure of the edited *scar-6* lncRNA (Figure EV2C- highlighted region). It is highly plausible that the disrupted stem-loop structure, caused by the 12 bp deletion, serves as a functional interacting motif. Its disruption likely leads to the observed coagulation phenotype. Our previous study on lncRNA *VEAL2* also have shown that structural change due to indels disrupts the function of the lncRNA.

Line 259-269

“The unchanged levels of *scar-6* lncRNA suggest that the 12-bp deletion may not affect its degradation but could potentially alter its secondary structure, thereby disrupting its functional role. Therefore, we examined the secondary structure of the lncRNA with and without the 12-bp deletion using RNAfold web server (Lorenz *et al*, 2011). We observed a significant alteration in the secondary structure of the lncRNA following the deletion of 12 bp (Fig EV2C). The observed changes, particularly in the hairpin loop structure, could profoundly impact the *scar-6* lncRNA's function. This structural alteration might affect its ability to interact with target molecules, regulate gene expression, and stabilize transcription factor binding at the regulatory site. These disruptions could render the *scar-6* lncRNA nonfunctional, explaining the observed changes in *prozb* expression.”

Figure EV2 : [C] Representative image showing secondary structure of scar-6 RNA under wildtype and 12-bp edited condition. The highlighted region represent change in the structure of lncRNA

5. Line 256 "We next evaluate the functional modality of the scar-6 lncRNA": rescue experiments using injections of in vitro transcribed scar-6 RNA would make sense only if scar6 lncRNA is degraded/downregulated in the 12-nt deletion mutant. As already mentioned above, the authors should test scar-6 expression in the mutant. Moreover, if the function of the scar-6 transcript indeed is dependent on the missing 12-nt, injections of mature scar-6 delta 12nt RNA should be used as a control for no rescue.

Ans: We did performed qRT-PCR to check the expression of the scar-6 lncRNA at 24 hpf and 72 hpf in the *scar-6^{gib007Δ12/Δ12}* zebrafish to address that there was no significant change in the *scar-6* RNA levels (figure 2J) but we do observe change in its secondary structure (Figure EV2C). We do agree with the reviewers point that rescue experiments using injections of in vitro transcribed *scar-6* RNA would make sense only if *scar-6* lncRNA is degraded/downregulated in the 12-nt deletion mutant. Additionally, we would like to highlight that rescue experiments are also conducted to restore the wild type structure functional copy of the *scar-6* lncRNA in the mutant animals. The secondary or tertiary structures of lncRNAs have been shown to play a crucial role in their interaction and functional capacity. In our manuscript, we provide evidence that a 12 bp deletion disrupts the secondary structure of the elncRNA (Figure EV2C) and is indeed responsible for the coagulation phenotype. We demonstrated that the coagulation defect in the mutant could be rescued by introducing a wild-type (WT) copy of the *scar-6* elncRNA. Additionally, as per reviewers suggestion we performed an experimental attempt to rescue the mutant with the 12 bp deleted elncRNA did not result in any significant changes in the coagulation phenotype. This strongly indicates that the *scar-6* elncRNA regulates coagulation in zebrafish, and that the 12 bp deletion adversely affects its expression and function.

Line 303-306

“We assessed the hemostatic ability of the mutants after rescue with either the wild-type or 12 bp deleted *scar-6* lncRNA IVT RNA. Our results showed that the in-crossed *scar-6^{gib007Δ12/+}* animals could restore hemostatic function with the wild-type copy of *scar-6*, but not with the 12 bp deleted copy (Fig 3F).”

Figure 3 [F]Coagulation assay plot calculating the time of occlusion (s) in different individual zebrafish in control, in-cross progeny of *scar-6*^{gib007Δ12/+}, in-cross progeny of *scar-6*^{gib007Δ12/+} injected with WT *scar-6* IVT RNA(100ng/ul) and in-cross progeny of *scar-6*^{gib007Δ12/+} injected with *scar-6*^{gib007Δ12} IVT RNA(100ng/ul) performed blinded. Each point representing the occlusion time of individual zebrafish segregated based on injected concentration; ns- not significant ** p<0.01 **** p<0.0001 (Mann-Whitney U)

6. Line 273: "...the RNA does not appear to have any discernible impact or independent function on the cranial hemorrhage morphology or cis-regulatory effect on *prozb*". The sentence is misleading. As the authors inject mature (spliced, capped, polyadenylated) *scar-6* RNA, cis-regulatory effect on *prozb* expression cannot be tested with this experiment. The cis regulatory effect of the *scar-6* could have been tested if *scar-6* was expressed from a DNA expression construct.

Ans: Thank you for raising this concern. We have revised the text for clarity. Here we want to convey that the *scar-6* elncRNA transcript it self cannot regulate the *prozb* levels. Late on in the manuscript we have demonstaracted that it also requires the enhancer present on *scar-6* locus to cis-regulate *prozb* function. We have changed the text in the manuscript for better clarity of the work.

Line 316-323

"These findings indicate that the RNA produced from the *scar-6* locus plays a role in regulating the hemostatic ability of zebrafish.

Taken together these observations suggests the RNA produced from the *scar-6* locus does not appear to impact cranial hemorrhage morphology or significant change in the transcriptional

regulation of the *prozb* gene in *scar-6*^{gib007Δ12/Δ12} animals. We speculate that the function of the *scar-6* lncRNA may be context-dependent in its role in controlling cranial hemorrhage and *prozb* regulation. Alternatively, other regulatory elements, possibly enhancer, might be involved in regulating *prozb* expression and cranial hemorrhage in the *scar-6* mutants.”

We agree with the reviewer’s perspective and have demonstrated that the regulation of *prozb* expression is mediated through DNA-based enhancer cis-regulation and enhancer-promoter looping of *scar-6/prozb*. Our rescue experiment was designed to test if the mature *scar-6* lncRNA has any major impact on the cis-regulation or not and hence we injected an fully preprocessed mature IVT RNA in the mutant. This is an standard practice in zebrafish filed to recue the altered gene. (Sehgal P ,et al. 2021, PMID: 34180064; Palasin K, et al. 2019;PMID: 31792295; Veleri S, et al. 2012 , PMID: 22479622; Xu X, et al. 2012, PMID: 22353712)

7. When overexpressing in vitro transcribed scar-6 RNA injected into the one-cell stage embryos, the authors measure time to occlusion at 3 days post fertilization (dpf). My concern is that the effect of RNA overexpression is scored at a very late developmental stage; it is unlikely that injected, in vitro transcribed RNA is stable until 3dpf. Same is valid for the following statement "We observed no prominent cardiovascular phenotype till 5dpf upon knockdown. This is a very late time point for a morpholino-induced phenotype to be evaluated.

Ans: Thank you for raising this concern. We appreciate your attention to detail. Indeed, the stability of in vitro transcribed (IVT) RNA is an important aspect to consider in such experiments. We acknowledge that the injected RNA might undergo degradation over time, and its stability is influenced by factors such as the half-life of the RNA. To address this concern, we did injected a substantial number of RNA copies into the one-cell stage embryos, ensuring that even after the expected half-life of the IVT RNA, a significant amount of functional RNA is available for the intended effects. This practice aligns with established standards in the field, and several published articles support the efficacy of this approach(Sehgal et al., 2019, PMID: 34180064; Sarangdhar MA et al. 2018, PMID: 30011017; Matharu NK et al 2021 PMID: 33994355). Moreover, we performed RT-PCR quantification of the injected IVT RNA at 3dpf to provide additional evidence. The data obtained from this analysis confirms that a substantial copy number of the injected RNA is still present, supporting the functional aspect of our study. We had already provide this data in Supplementary Figure 9A which is represented below.

Supplementary Figures 9:[A] Relative fold change expression of *scar-6* at 3dpf upon *scar-6* IVT RNA overexpression in zebrafish at different concentrations (100, 200 500 ng/uL). Data from 3 independent biological replicates plotted as mean fold change \pm standard deviation. ** $P < 0.01$, *** $p < 0.001$, **** $p < 0.0001$ (two-tailed unpaired t-test).

We used morpholinos to knock down the *scar-6* transcript in zebrafish. The efficacy of morpholinos has been estimated to last approximately 5–6 days post-fertilization (dpf) in zebrafish maintained at a standard temperature of 28 °C, as supported by multiple studies (e.g., Giacomotto J, et al. 2015, PMID: 26051838). To ensure we did not miss any phenotypic effects due to later-stage knockdown, we monitored the embryos for defects up to 5 dpf. We also performed qRT-PCR to validate the knockdown of the *scar-6* transcript in zebrafish at 3dpf where the functional assay was performed (Figure EV 3B). We hope this clarifies the experimental design and addresses your concern.

Figure EV3 [B] Relative fold change expression of *scar-6* upon of knockdown with splice blocking morpholino in zebrafish at 3dpf. Data from 3 independent biological replicates plotted as mean fold change \pm standard deviation; * $P < 0.05$ (two-tailed unpaired t-test)

Referee #2:

The manuscript by Gyan Ranjan et al. is about a previously uncharacterized lncRNA in zebrafish, which they named *scar6*. The lncRNA originates from a complex locus and partially overlaps with the gene *f10* (Coagulation Factor X). The authors identified *scar6* through syntenic analysis of lncRNA genes associated with the cardiovascular system. They describe that KO or deletion of 12nt of *scar6* has severe physiological effects (mainly vascular; decreased survival, longer occlusion time, hemorrhage defects). Some effects could be rescued by injection of an in vitro transcribed version of *scar6*. Molecularly, the $\Delta 12$ *scar6* had no significant effects on *f10* or *scar6* expression itself, however, it led to an upregulation of the neighboring gene *prozb*. The authors suggest that *prdm14* binds *scar6* RNA and genomic DNA, recruiting PRC2-subunit *Suz12* to repress the expression of the lncRNA. This allows CTCF to induce *prozb* expression. *Prozb* activates the PAR2 system to induce in an NFkB-dependent manner the expression of cell adhesion molecules, which trigger cardiovascular dysfunction.

The manuscript is presented in a clear way, with minor language issues, and with a lot of data. The data clearly represents physiological importance of the molecules investigated. Also the molecular part was investigated. However, the highly complex locus of the lncRNA *scar6* as well as the translation towards human cell lines/types remains difficult.

This is the first description of lncRNA *scar6*. I am not 100% convinced that this lncRNA really exists. Therefore, it is not clear, how long the RNA is and where does it exactly start and stop. Additionally, the lncRNA overlaps with *F10*, which itself has important functions for the cardiovascular system. It is therefore absolutely crucial to demonstrate that the effects seen are really from *scar6* and not from a *f10* mutant.

Major points:

1. As this is a novel lncRNA, several experiments are needed. The authors did some initial experiments, however, a full 5' and 3' RACE is needed incl Sanger sequencing. This would allow to see the start and end of the transcript. Additionally, it should be experimentally checked how long the transcript really is. For this, a northern blot would be greatly appreciated. Also key experiments should be performed with an exon-spanning primer. For me, when I blast the primer sequences provided, some bind also the overlapping gene *f10*, questioning whether the primers are really specific (especially 15/16/3/6).

Ans: We did provide evidence that the transcript is true in nature. We performed 3'RACE for our *scar-6* lncRNA in zebrafish and observed it to show a similar 3' end as described in the ZFLNC database. We performed the 3'RACE and did sanger sequencing of the band observed. The BLAT alignment of Sanger sequencing result of *scar-6* is shown in the UCSC genome browser screenshot (line 192-194) (Figure 1B)(Supp Fig 2). We note that the expression levels of *scar-6* are likely too low for it to be robustly detected by a Northern blot, which has limited sensitivity. We did also try 5' RACE, but it was difficult to design primers for it as it has several AT repeat regions. Also 5' RACE for antisense transcripts are tricky as they overlap with protein coding gene and chances of false discovery is high. We also did strand specific amplification to prove that the transcript is antisense to the *f10* and it also provide clear demarcation of the exon-intron boundaries of the *scar-6* lncRNA.

Figure 1 B UCSC genome browser snapshot of *scar-6* locus in zebrafish. Strand-specific PCR and 3'RACE confirmed the transcript to be antisense to *f10* in zebrafish.

The primers 15/16/6 were specifically designed to be strand-specific for *scar-6*, ensuring they only amplify the *scar-6* locus. Additionally, RNA samples were treated with DNase to eliminate any residual DNA, confirming that the signal detected originates solely from strand specific RNA. Primer 3 represents the sgRNA sequence targeting the *scar-6* locus; as it targets DNA, it operates independently of the strand. We recognize that this could potentially disrupt the *f10* gene. To address this, we conducted additional experiments to verify that the observed phenotypes are indeed due to the disruption of the *scar-6* lncRNA and have provided it in the Supplementary Figure 10.

2. It is not clear to me, why exactly these 12nt were selected for the mutation and not some other 12nt of scar6. Please explain.

Ans: In our lab, we have generated multiple lncRNA knockouts in zebrafish using minimally invasive genome editing techniques. We have observed prominent phenotypes resulting from small indels. One of our studies, published in The EMBO Journal, demonstrated that an 8 bp deletion can significantly disrupt *VEAL2* lncRNA function in zebrafish (Sehgal P ,et al 2021, PMID: 34180064;). Additionally, several studies have shown that specific motif deletions, such as those in cyrano, can result in observable phenotypes (Goudarzi et al., 2019; PMID: 30620332). Based on this and our previous experience, we used a strategy to target both exons of the lncRNA. We anticipate that indel mutations will disrupt the secondary or functional structure of the lncRNA, rendering it non-functional. We also used F₀ screening strategy to select our mutant. Our previous observations, supported by multiple studies, indicate that CRISPR-Cas9 mediated editing and phenotypic screening in zebrafish can be efficiently conducted at the F₀ generation (Kroll F,et al 2021, PMID: 33416493; Lu F et al 2023 PMID: 36823429; Deniz E, et al 2018 PMID: 30151766; Parvez S et al 2023 PMID: 37069311 . Generation of stable mutants in zebrafish is financial and time consuming. This approach has proven to be an effective strategy for selecting mutants, thereby reducing both time and resources required for subsequent generations.

In the current study, we systematically designed two sgRNAs for our target *scar-6* lncRNA gene. One sgRNA targeted the 5' region (exon1) of the lncRNA gene, but F₀ screening of these animals injected with CRISPR-Cas9 RNP complex resulted in no substantial phenotypic changes. Conversely, the sgRNA targeting the 3' region (exon2) of the lncRNA gene exhibited a pronounced hemorrhage phenotype during our initial F₀ screening, prompting us to pursue this

lead. We also identified other mutants in the 3' region, but they did not consistently show the hemorrhage phenotype. Therefore, we focused on the 12 bp deletion mutants for our study, as this was the largest deletion observed upon F₁ screening and it consistently exhibited the hemorrhage phenotype across generations. This initial F₀ screening emphasizes the specific and pivotal role of the 12bp deletion in influencing the observed phenotype and underscoring its functional importance in the context of our investigation. This is represented below and is given in the revised manuscript Supplementary Figure 6.

Supplementary Figure 6: - Representative image showing transgenic *gib004Tg(fli1a:EGFP;gata1a:DsRed)* 3dpf zebrafish injected with RNP complex of CRISPR-Cas9 targeting 5' and 3' region of *scar-6* gene (5x resolution)

3. Since the 12nt deletion of *scar6* potentially overlaps with *f10*, it should be demonstrated that it is really the lncRNA mutation mediating this effect. This could be done by injecting a IVT of *f10* (the overlapping part; so *scar6* antisense) and testings for rescue.

Ans:- Thank you for your insightful comment and concern. We acknowledge the reviewer's point, and we have already provided evidence that the *f10* is not the cause of the phenotype in the manuscript accordingly in Supplementary figure 10.

- We performed qRT-PCR to address that there was no significant change in the *f10* mRNA levels (figure 2J). (N- 3 biological replicate)

Figure 2 [J]Bar plot representing relative fold change in expression levels of *f10*, *scar-6* and *prozb* between wild-type and *scar-6*^{gib007Δ12/Δ12} zebrafish. The *prozb* expression levels were 8-fold increase in the *scar-6*^{gib007Δ12/Δ12} zebrafish compared to the wild-type. Data from 3 independent biological replicates plotted as mean fold change ± standard deviation; *** p-value ≤ 0.001 (unpaired two-tail t-test)

- We also performed western blotting for *f10* protein in the wildtype and *scar-6*^{gib007Δ12/Δ12} zebrafish animals and found no change in the protein levels.(Supp Fig 9A)
- We also performed a rescue experiment by injecting *f10* mRNA IVT (100ng/uL) in the single cell of *scar-6*^{gib007Δ12/+} incrossed progeny and found no rescue in the hemorrhage phenotype of the animals (supp Fig 9 B-C).

Supplementary Figures 9: Rescue of *scar-6* mutant with *f10* mRNA

[A]Western blot of *f10* in wild type and *scar-6*^{gib007Δ12/Δ12} mutant zebrafish. The bar plot represents the quantification of the western blot from 3 independent biological replicates plotted as mean fold change ± standard deviation.ns- not significant (two-tailed unpaired t-test)

[B]Box plot representing the percentage of animals exhibiting hemorrhage phenotype in wild type and *scar-6*^{gib007Δ12/Δ12} mutant zebrafish injected with *f10* IVT RNA (100 ng/uL).Data from 3 independent biological replicates plotted as mean percentage ± standard deviation; *** p<0.001, **** p<0.0001 (two-tailed unpaired t-test).

[C]Representative image showing the cranial region of 3dpf zebrafish with o-dianisidine staining of RBC blood cells in wild type and *scar-6*^{gib007Δ12/Δ12} mutant zebrafish injected with *f10* IVT RNA (100 ng/uL). (4x magnification)

Additionally, the data from the previous *f10* knockout study (Hu,z *et.al*; 2017) demonstrates loss of mRNA and phenotype in later adult stages of zebrafish. In contrast, our study reveals the phenotype in early zebrafish stages, contradicting the expected outcome based on observations

from Hu et al., 2017. Collectively it suggests that the phenotype is due to *scar-6* elncRNA gene disruption and there is no major impact of the 12 bp distribution on *f10* expression and function.

4. Since the authors described a potential homolog of *scar6* in humans, the relevance of this homologue should be tested. When i checked RNA-seq data in HUVEC, SCAR6 and the other genes werent expressed, questioning the translational relevance of the findings. However, the authors should investigate in primary cells (not HUVEC/hTERT) from the cardiovascular system (e.g. HUVEC, smooth muscle cells, cardiomyocytes or similar) how highly SCAR6 is expressed and whether the knockdown also upregulate PROZB.

Ans:- Thank you for your insightful comment and concern. The human *SCAR-6* elncRNA is expressed at a very low level as observed in zebrafish, necessitating high-coverage RNA sequencing of HUVEC cells for detection. The human *SCAR-6* transcript is independently annotated by GENCODE, ENSEMBL, and FANTOM, further validating its authenticity. Additionally, ENCODE data confirms its expression in the HepG2 cell line as described below in the UCSC browser screenshot (It also exhibits CAGE reads in antisense direction providing evidence that its an antisense RNA), supporting the genuine nature of this transcript. Previous reports have suggested that coagulation cascade genes are highly expressed in liver tissue and are secreted from it. Given that *SCAR-6* is known to regulate genes in the coagulation cascade, its expression is expected to be in the liver or hepatocytes, which we have also observed. Under normal conditions, we do not expect to see higher expression of these genes in HUVEC cells, as their overexpression has been linked to pathological phenotypes as reported by multiple studies (Rabiet MJ, et al 1996, PMID: 8630677; Bono F et al 2011, PMID: 11073863.; Noll T et al 1999 PMC2193057).

Figure for reviewers : UCSC screenshot of SCAR-6 locus (ENSG00000283828) with tracks of HepG2 transcript reads from ENCODE data and CAGE reads suggesting antisense transcript (Red +ve strand; Blue -ve strand).and Conservation scores. (This data is not in manuscript)

5. The authors showed that the 12nt deletion of *scar6* led to an upregulation of *prozb*. What about the other genes in that locus (F7, PCID2, MCF2L)? Are they also affected? Is *prozb* the highest induced gene from the mutation - did the authors do a RNA-seq from the *scar6* mutant?

Ans:- We did analyze the expression of neighboring genes using qRT-PCR and observed no significant changes. (MCF2L is not conserved in synteny with zebrafish.)

Due to financial constraints and our primary focus on elucidating the role of *scar-6* in regulating its cis-genes, we did not conduct RNA-seq for the *scar-6* mutant. However, RNA-seq would be advantageous for identifying potential trans-regulatory effects or other genes impacted by the *scar-6* mutation. This is an ongoing area of investigation and is not part of the current study.

6. The induction of *prozb* was only shown on RNA level. Please provide an evidence on protein level (western blot).

Ans: Despite testing several *PROZ* human polyclonal antibodies, none exhibited reactivity with zebrafish lysates and there was no available zebrafish-specific *prozb* antibody commercially available. Consequently, we faced limitations in blotting the *prozb* protein in our mutant. However, our qRT-PCR and overexpression of *prozb* data emphasize that the elevated *prozb* levels are the underlying cause of the observed hemorrhage phenotype.

7. Why should *prdm14* bind both, the RNA and the genomic DNA of *scar6*? As the genomic DNA binding and recruitment of PRC2 seems a logical explanation, the RNA binding of *prdm14* seems contradictory as *scar6* RNA would rather implicate that a different function.

Ans: A recent publication by Dr. Richard A. Young's group (Oksuz O, et al., 2023, PMID: 37402367) suggests that at least half of transcription factors (TFs) also bind RNA. This occurs through a previously unrecognized domain with sequence and functional features similar to the arginine-rich motif of the HIV transcriptional activator Tat. This RNA binding enhances TF function by promoting dynamic associations between DNA, RNA, and TFs on chromatin. Additionally, transcription at active regulatory elements may create a positive feedback loop that stabilizes regulatory elements, thereby reinforcing gene expression programs (Sigova AA, 2015, PMID: 26516199). Numerous studies, including those on γ -globin/BGLT3 (Ivaldi MS et al 2018, PMID: 30150205), E2/ER- α /eRNA (Li W, et al 2013 PMID: 23728302), AR/KLK3 eRNA (Hsieh CL et al 2014 PMID: 24778216), Cga/eRNA (Pnueli L, et al 2015 PMID: 25810254), and INK4a/ARF (Farooq U, et al 2021 PMID: 33761351), have shown that eRNAs produced from enhancer loci can aid in chromatin remodeling and the stability of TFs at these loci. Therefore, it is plausible that *scar-6* eRNA binding with *prdm14* may enhance its binding stability at DNA regulatory regions, a hypothesis we have experimentally validated (Figure 5E). Additionally, we cannot rule out other functions of *scar-6* eRNA, as studies of various lncRNAs have demonstrated their potential to perform multiple roles and we do agree a more detailed study needs to be performed to elucidate its other functions.

Minor points:

1. Introduction: Lines 61-65. I disagree with the statement that only a small subset of lncRNAs has been studied. The literature is enormous and since years every day new studies are published. Therefore i would recommend to reconsider this statement.

Ans: Thank you for pointing it out. We have modified the sentence. We were pointing out to the vast majority of lncRNA (~50,000 lncRNA transcripts) which are still uncharacterised and less than 1% of it has only been characterized for function.

Line 61-62

“Long non-coding RNAs (lncRNAs) has been investigated for their functions and has been found to regulate protein-coding genes in *cis* or *trans* for various biological conditions”

2. Introduction: Additionally, it should be mentioned that lncRNAs are physiologically important (see e.g. DOI: 10.1007/s00424-021-02641-z).

Ans: Thank you for the suggestion to improve our manuscript. As suggested we have incorporated the physiological reliance of lncRNA in the introduction.

Line 66-72

“Numerous lncRNAs, including *MALAT1*, *Caren*, *MANTIS*, *Mhrt*, *Tug1*, and *Pnky*, have been shown to be physiologically crucial for organ development, maturation, and function (Cremer *et al*, 2019; Sato *et al*, 2021; Leisegang *et al*, 2017; Lewandowski *et al*, 2020; Ramos *et al*, 2015). These lncRNAs provide compelling evidence of their significant roles in regulating key biological processes, underscoring their importance in maintaining proper organ physiology (Oo *et al*, 2022)”

3. Introduction: Also the different ways how lncRNA can act should be mentioned (see e.g. DOI: 10.1038/s41467-024-45183-5)

Ans: Thank you for suggestion to improve our manuscript. As suggested we have incorporated the physiological reliance of lncRNA in the introduction.

Line 92-100

“lncRNA loci are categorized based on their mechanisms of action. Some function through their mature RNA molecules, interacting with proteins, RNA, or DNA to regulate cellular processes (e.g., *VEAL2*, *GATA6-AS*, *Tie1-AS*, *lncREST*) (Sehgal *et al*, 2021; Neumann *et al*, 2018; Li *et al*, 2010; Statello *et al*, 2024). Others act through the transcription process, serving as bidirectional promoters or transcription and splicing regulators (e.g., *ARIN*, *Handsdawn*) (Santoro *et al*, 2013; Ritter *et al*, 2019). Additionally, some loci operate exclusively through their DNA elements, functioning as chromatin, enhancer, or transcription factor binding regions (e.g., *PVT1*, *linc-p21*, *Lockd*, *Meteor*) (Cho *et al*, 2018; Groff *et al*, 2016; Rom *et al*, 2019; Gil *et al*, 2023).”

4. Material and methods: 567-573: Which method was used to analyze RT-qPCR data?

Ans: Thank you for your suggestions and comments. We have provided additional text describing the method used for RT-qPCR analysis.

line 679-680

“The relative levels of specific transcripts in the original pool of RNA were estimated using the methods described (Winer *et al.*, 1999; Livak and Schmittgen, 2001)”

5. Material and methods: The buffers used for several experiments are not listed in detail. What ingredients do they have? E.g. a hypotonic buffer was used for fractionation - this is

not helpful and readers might have severe problems to validate their findings.

Ans: Thank you for your suggestion and concern. We have modified the text and provide full description of the buffer used. RIPA and NP-40 lysis buffer were commercially ordered from Thermo scientific and it has been mentioned in the material and method.

Line 709-711

The single cells were incubated and homogenized in hypotonic buffer (10mM HEPES, Ph 7.9; 1.5mM MgCl₂; 10mM KCl; 0.5mM DTT;and Protease inhibitor) followed by centrifugation at 1000 rpm at 4°C for 5 min.

Line 736

PNK buffer (10mM NaCl, 10mM Tri-Cl pH7.6, 1mM EDTA, 0.5% SDS)

6. Maybe i did not find it, but the dCas System used is not explained in the methods.

Microinjection of morpholino and dCas9-KRAB into the embryo.

Ans: Thank you for pointing it out in the comment. We apologize for missing it in the material and methods section. We have added the section in the manuscript.

Line 645-657

“Antisense splice blocking morpholino oligonucleotide (MO) was deside targeting the 2nd exon of scar-6 (GeneTools, USA) (supplementary table 4). Morpholino was dissolved in nucleus free water to a final stock concentration of 1mM. 3nl of different concentrations of MO was injected into 1 cell stage of double transgenic zebrafish gib004Tg(fli1: EGFP; gata1a: dsRed) and the phenotype was observed. Scrambled MO (CCTCTTACCTCAGTTACAATTTATA) was used for control injection.

Knockdown using dCas9-KRAB was performed as previously described (Long et al, 2015). We generated in vitro transcribed (IVT) RNA of the amplified dCas9-KRAB region from the Lenti-(BB)-EF1a-KRAB-dCas9-P2A-EGFP plasmid (a gift from Jorge Ferrer; Addgene plasmid #118156; <http://n2t.net/addgene:118156>; RRID) using the T7 mMessage mMachine kit. The sgRNA used in the knockout experiment was used. The injection was done at a final concentration of 300 ng/μl for dCas9-KRAB and 100 ng/μl for sgRNA. A single injection of dCas9-KRAB IVT was used as a control.”

7. The plasmids/vectors used in the manuscript are not explained in the material and methods. Please also indicate when they come from Addgene.

Ans: Thank you for pointing it out in the comment. We have provided the details of plasmid used in the study and their addgene description as described by addgene website.

Line 651-654

“We generated in vitro transcribed (IVT) RNA of the amplified dCas9-KRAB region from the Lenti-(BB)-EF1a-KRAB-dCas9-P2A-EGFP plasmid (a gift from Jorge Ferrer; Addgene plasmid #118156; <http://n2t.net/addgene:118156>; RRID)”

Line 684-685

“An enhancer assay was performed using E1b-GFP-Tol2 (a gift from Nadav Ahituv; Addgene plasmid # 37845 ; <http://n2t.net/addgene:37845> ; RRID:Addgene_37845).”

8. Please indicate also which software or program was used to analyze TFs, synteny, non-coding probability.

Ans: Thank you for pointing it out in the comment. We have added the section in the materials and method section. The synteny conservation of lncRNAs between the zebrafish and human was obtained from ZFNC database.

Line 590-594

Using the ZFLNC database (<https://www.biochen.org/zflnc/>), we curated out the list of lncRNAs that were conserved between zebrafish and humans. In the database, 3 modes of conservation were used to identify the conserved lncRNA. For our study, we selected the syntenic and collinearity model of conservation and retained those lncRNA transcripts.

Line 605-607

We also computed the non-coding potential of the lncRNAs using bioinformatic algorithms such as the Coding Potential Assessment Tool (CPAT2) (Wang *et al*, 2013) or Coding Potential Calculator (CPC2)(Kang *et al*, 2017).

Line 612-614

Transcription factor binding at the scar-6 locus was predicted using the JASPAR 2022 database (Castro-Mondragon *et al*, 2022) plugin track in the UCSC genome browser (Raney *et al*, 2024) for zebrafish danRer11 reference genome.

9. Which NGS tracks were used to show the genomic loci, their epigenetic environment and repetitive sequences in zebrafish? This must be mentioned in the material and methods. If e.g. UCSC was used, also this has to be cited.

Ans: Thank you for pointing it out in the comment. We have added the section in the materials and method section.

Line 615-617

Additionally, the UCSC genome browser (Raney *et al*, 2024) track for the zebrafish danRer10 genome was utilized to visualize genomic locations, repeat regions, and epigenetic marks (Danio-code data; (Raney *et al*, 2024; Baranasic *et al*, 2022)).

Dear Dr. Sivasubbu,

Thank you for the submission of your revised manuscript. We have now received the enclosed report from referee 2 whom I asked to assess how well the concerns to all referees were addressed, given that referee 1 was not available for re-review. I am happy to say that referee 2 supports the publication of your ms now. Only a few editorial requests will need to be addressed before we can proceed with the official acceptance of your manuscript:

- Please submit a ms word file without figures with your final submission.
- Please reduce the number of keywords to 5.
- Please correct the conflict of interest subheading to "Disclosure and Competing Interests Statement"
- We need a functional institutional email address for all corresponding authors.
- The author credits need to be removed from the ms file. All credits need to be entered during online ms submission.
- Please submit with your final ms a completed author checklist, which you can download from our author guidelines <<https://www.embopress.org/page/journal/14693178/authorguide>>. The completed author checklist will also be part of the transparent peer-review process file.
- The funding information needs to be part of the Acknowledgments section.
- Please upload all main and all EV figures as individual, high resolution figure files.
- Fig 6G is called out but not labeled; callouts for Supp Fig 10, 11 and 17 are missing, please correct/add.
- The Supplementary Data in the ms need to be removed and uploaded separately as a PDF file titled Appendix; the Appendix file needs a table of Content on the title page with page numbers to show where each item is located; the correct nomenclature and ms callouts for all the items is Appendix Table S1-S4 and Appendix Figure S1-S17. You can also chose the 5 most important Appendix figures as EV figures that will be embedded in the ms text online. Please check our guide for authors online for more information.
- Since July this year, we ask authors to upload a reagents and tools table with their ms. You can download a template (.docx) for the Reagents and Tools Table from our author guidelines: <<https://www.embopress.org/page/journal/14693178/authorguide#manuscriptpreparation>>.
- The manuscript sections should be in the following order: Title page - Abstract & Keywords - Introduction - Results - Discussion - Methods - Data Availability - Acknowledgments - Disclosure Statement & Competing Interests - References - Figure Legends - (Main Tables with legends) - Expanded View Figure Legends.
- In our figure check of ms, we detected a few image re-uses:
 1. Image re-use between Figure 3G middle image and Supp Figure 6 - bottom right - N= 24/79
 2. Image re-use between Figure 3E and Supp Fig 8
 3. Blot re-use between Figure EV2A and Supp Figure 7BPlease explain what happened. If the re-use is on purpose, please mention this in the figure legends.
- Please note that the figure panel 6g is not labeled in the manuscript, however the corresponding legend is labelled as 6g. This needs to be rectified.
- Please define the annotated p values ****/***/** as well as provide the exact p-values for the same in the legend of figure 2g; 6b; as appropriate.
- Please note that the exact p values are not provided in the legends of figures 2i-j; 3a-c, f, i; 4d-e; 5b, d-e; 6c-f; EV 3b, f; EV 4c; EV 5c.
- Please indicate the statistical test used for data analysis in the legends of figures 2g; 6b.
- Please note that in figure 3i; there is a mismatch between the annotated p values in the figure legend and the annotated p values in the figure file that should be corrected.

- Please note that the box plots need to be defined in terms of minima, maxima, centre, bounds of box and whiskers, and percentile in the legend of figure EV 1e.
- Please note that the box plots need to be defined in terms of bounds of box and whiskers, and percentile in the legend of figure EV 3c.
- Please note that the box plots need to be defined in terms of minima, maxima, centre, bounds of box and percentile in the legend of figure EV 4c.
- Please note that information related to n is missing in the legends of figures 3c; EV 1e.
- Please note that the error bars are not defined in the legends of figures 2e, g; 3h
- Please note that the scale bar is missing for figures 1f; 2f; EV 3d; EV 4b.
- Please note that scale bar and its definition are missing for figures 2h; 3g; 4c; 6a; EV 2b.
- Please note that the red asterisk is not defined in the legend of figure EV 2a. This needs to be rectified.
- Please note that the black/white/red arrowheads are not defined in the legend of figure 1f; 2f; 4c; 6a; EV 3d; EV 4b. This needs to be rectified.

I would like to suggest a few changes to the abstract that needs to be written in present tense. Do you agree with the following:

In this study, we characterize the lncRNA locus known as Syntenic Cardiovascular Conserved Region-Associated lncRNA-6 (scar-6) and functionally validate its role in coagulation and cardiovascular function. Partial deletion of the scar-6 locus in zebrafish (scar-6gib007 Δ 12/ Δ 12) results in cranial hemorrhage and vascular permeability. Both overexpression, knockout and rescue of scar-6 lncRNA modulates hemostasis in zebrafish. Molecular investigation reveals that the scar-6 RNA acts as an enhancer lncRNA (elncRNA), and controls the expression of prozb, an inhibitor of factor Xa, through the enhancer element on its locus. The scar-6 locus suppresses loop formation between prozb and scar-6 sequences, facilitated by the methylation of CpG islands via the prdm14-PRC2 complex that is stabilized by the scar-6 elncRNA transcript. This complex is disrupted in scar-6gib007 Δ 12/ Δ 12 zebrafish. Furthermore, activation of the PAR2 receptor in scar-6gib007 Δ 12/ Δ 12 zebrafish triggers NF- κ B-mediated endothelial cell activation, leading to vascular dysfunction and hemorrhage. We present evidence that the scar-6 locus plays a role in regulating the expression of the coagulation cascade gene prozb and maintains vascular (OK?) homeostasis.

EMBO press papers are accompanied online by A) a short (1-2 sentences) summary of the findings and their significance, B) 2-3 bullet points highlighting key results and C) a synopsis image that is exactly 550 pixels wide and 200-600 pixels high (the height is variable). The synopsis image should provide a sketch of the major findings, like a graphical abstract. Please note that text needs to be readable at the final size. Please send us this information as 2 individual files along with the final manuscript.

Referee #2:

The authors have satisfactorily answered all concerns.

I appreciate the efforts with RACE, and I know the difficulties about 5'RACE. For the expression in HUVEC, I found expression of the scar6 homologue in our 150mio deep RNA-Seq next to F10 as indicated by the authors. Also the CAGE is convincing. The change of the title and focus towards the scar6 locus is reasonable.

All editorial and formatting issues were resolved by the authors.

Dr. Sridhar Sivasubbu
Institute of Genomics and Integrative Biology
Mall Road
Delhi, Delhi 110007
India

Dear Dr. Sivasubbu,

I am very pleased to accept your manuscript for publication in the next available issue of EMBO reports. Thank you for your contribution to our journal.

You qualify for financial assistance for your publication charges - either via a Springer Nature fully open access agreement or an EMBO initiative. Check your eligibility: <https://www.embopress.org/page/journal/14693178/authorguide#chargesguide>
